

# Performance evaluation of three bio-optical models in aerosol and ocean color joint retrievals

Neranga K. Hannadige[1], Peng-Wang Zhai[1], Meng Gao[2,3], Yongxiang Hu[4], P. Jeremy Werdell[2], Kirk Knobelspiesse[2], and Brian Cairns[5]

[1]Joint Center for Earth Systems Technology, Department of Physics, University of Maryland Baltimore County, 1000 Hilltop Circle, Baltimore, MD 21250, USA
[2]NASA Goddard Space Flight Center, Code 616, Greenbelt, MD 20771, USA
[3]Science Systems and Applications, Inc., Greenbelt, MD, USA
[4]NASA Langley Research Center, 1 Nasa Dr, Hampton, VA 23666, USA
[5]NASA Goddard Institute for Space Studies, 2880 Broadway, New York, NY 10025

**Correspondence:** Pengwang Zhai (pwzhai@umbc.edu)

**Abstract.** Multi-angle polarimeters (MAP) are powerful instruments to perform remote sensing of the environment. Joint retrieval algorithms of aerosols and ocean color have been developed to extract the rich information content of MAPs. These are optimization algorithms that fit the sensor measurements with forward models, which include radiative transfer simulations of the coupled atmosphere and ocean systems (CAOS) based on adjustable atmosphere and ocean properties. The forward model

consists of sub-models to represent the optics of the atmosphere, ocean water surface, and ocean body. The representativeness of these models for observed scenes is important for retrieval success. In this study, we have evaluated the impact on MAP retrieval accuracy of three different ocean bio-optical models with 1, 3, and 7 optimization parameters that represent the spectral variation of inherent optical properties ($IOP(\lambda)$s) of the water body. The Multi-Angular Polarimetric Ocean coLor (MAPOL) joint retrieval algorithm was used to process data from the airborne Research Scanning Polarimeter (RSP) instrument

acquired in different field campaigns. We performed ensemble retrievals along three RSP legs to evaluate the applicability of bio-optical models along geographically varying waters. The average differences between the MAPOL aerosol optical depth (AOD) and spectral remote sensing reflectance ($R_{rs}(\lambda)$) retrievals and the MODerate resolution Imaging Spectroradiometer (MODIS) products are also reported. We studied the distribution of retrieval cost function values obtained for the ensemble retrievals using the 3 bio-optical models under clear to highly turbid waters. For the 1-parameter model, retrieval cost function

values show narrow distributions over any type of water, regardless of the cost function values, whereas for the 3 and 7-parameter models, the retrieval cost function distribution is water type dependent, showing the widest distribution over clear, open waters. We observed that the 3 and 7-parameter models have similar MAP retrieval performances relative to the 1-parameter model. We also demonstrated that the 3 and 7-parameter bio-optical models can be used to accurately represent both clear, open, and turbid, coastal waters, whereas the 1-parameter model is most successful over extremely clear waters. Given

the computational efficiency requirements, we recommend the 3-parameter bio-optical model as the coastal water bio-optical model for future MAPOL studies. This study guides MAP algorithm development for current and future satellite missions



such as NASA's Plankton, Aerosol, Cloud, ocean Ecosystem (PACE) mission and ESA's Meteorological Operational-Second Generation (MetOp-SG) mission.

## 1  Introduction

The enhanced capabilities in satellite remote sensing of Earth have enabled detailed observation of the atmosphere, ocean, and land thereby improving the accurate determination of spatial and temporal distributions of the constituents of each. Satellite-borne spectroradiometers in particular have substantially advanced the way we view our home planet, and their information content will increase in the future as the technology evolves from multi- to hyperspectral capabilities. Multi-angle polarimeters (MAPs), such as the POLarization and Directionality of the Earth's Reflectance (POLDER) (Deschamps et al., 1994), Air-borne Multi-angle Spectro-Polarimetric Imager (AirMSPI) (Diner et al., 2013), Spectro-polarimeter for Planetary Exploration (SPEX) (Smit et al., 2019), Research Scanning Polarimeter (RSP) (Cairns et al., 2003), Multi-viewing Multichannel Multi-polarization Imager (3MI) (Fougnie et al., 2018) and Multi-Angle Imager for Aerosols (MAIA) (Van Harten et al., 2021) have even greater information content compared to other existing single viewing angle spectroradiometers, such as the MODerate resolution Imaging Spectrometer (MODIS), Visible Infrared Imaging Radiometer System (VIIRS), and Ocean and Land Colour Instrument (OLCI), owing to their ability to perform measurements at multiple viewing angles and different polarimetric states (Dubovik et al., 2019).

Atmospheric aerosols play a critical role in Earth's climate and air quality (Boucher et al., 2013; Li et al., 2017). Aerosols affect Earth's energy balance directly by absorbing and scattering solar radiation and indirectly by interacting with clouds. Some of the traditional retrieval algorithms such as those for MODIS-like instruments result in larger aerosol and ocean color retrieval uncertainties (Remer et al., 2005; Sayer et al., 2016) when compared with the accuracy required for climate modeling (Mishchenko et al., 2004). The resultant uncertainties limit the accuracy of aerosol radiative forcing determination, thereby hindering our understanding of global climate change (Boucher et al., 2013).

Improved aerosol characterization and quantification will support accurate estimation of atmospheric path radiance in the atmospheric correction (AC) process (Mobley et al., 2016) of ocean color remote sensing. AC is the process of removing atmospheric and surface contributions from the total measured signal at the top of the atmosphere (TOA), so that ocean color can be assessed. The spectral remote sensing reflectance ($R_{rs}(\lambda)$ [$sr^{-1}$]) estimated through the AC process can be used to infer ocean optical and biogeochemical properties that are important for a broader understanding of phytoplankton dynamics, primary production, global carbon cycle and ocean's ecological response to climate change (Frouin et al., 2019). Accurate estimation of aerosols is thus important for both aerosol and ocean color retrievals.

AC algorithms can be divided into two categories of processing strategies. Heritage AC algorithms (Gordon and Wang, 1994) applied to MODIS-like spectroradiometers assume the atmosphere and ocean as decoupled entities with a single scattering





approximation (Zibordi et al., 2009; Gordon, 2021). The water leaving radiance in the near-infrared (NIR) is assumed to be negligible or appropriately modeled (the so-called black pixel assumption) (Bailey et al., 2010) to estimate the aerosol path

radiance at visible wavelengths by extrapolating from the NIR. This assumption does not unequivocally work in optically complex water. Furthermore, this can lead to an overestimate of aerosol path radiance with either nonzero NIR water leaving radiance or when absorbing aerosols are present (IOCCG, 2000). These approximations and assumptions make the heritage algorithm most applicable over clear open waters and clean aerosol conditions (IOCCG, 2000, 2010). The heritage algorithm implemented by NASA's Ocean Biology Processing Group (OBPG; https://oceancolor.gsfc.nasa.gov) works well over the open

waters but can produce negative $R_{rs}(\lambda)$ in blue wavelengths over turbid waters (Bailey et al., 2010). Efforts have been made to overcome negative $R_{rs}(\lambda)$ (Bailey et al., 2010; He et al., 2012; Fan et al., 2021; Ibrahim et al., 2019). None of the improvements have fully resolved the problem.

    The second category of AC algorithms makes use of the larger information content available from MAPs. These instruments have a greater capability to characterize aerosol microphysical properties (Mishchenko and Travis, 1997; Chowdhary et al.,

2001; Hasekamp and Landgraf, 2007; Knobelspiesse et al., 2012; Remer et al., 2019a, b) and as such, offer the potential for improvements in both aerosol and ocean color retrievals. Joint retrieval algorithms for MAPs have been developed, which provide simultaneous retrievals of aerosols and ocean color over both open and coastal waters (Chowdhary et al., 2005; Hasekamp et al., 2011; Xu et al., 2016; Stamnes et al., 2018; Gao et al., 2018, 2019, 2020, 2021; Fan et al., 2021).

    Joint retrieval algorithms fit the sensor measurements with forward model simulations for the coupled atmosphere and ocean

system (CAOS). The simulations are carried out by vector radiative transfer models with parameterizations that define the state of the CAOS. The difference between measurements and the model simulation is quantified by a cost function, which is minimized by iteratively perturbing the free parameters in the radiative transfer model. The forward model of ocean color joint retrieval algorithms consists of sub-models to simulate the optics of the CAOS, which is composed of the atmosphere, ocean surface, and ocean body. Like heritage approaches, the robustness of the aerosol and ocean retrievals from joint retrieval

algorithms depends on the representativeness of CAOS models over an observed scene. One important component of the forward model of ocean color joint retrieval algorithms is the suite of ocean bio-optical models that represent the spectral behaviors of aquatic inherent optical properties (IOP($\lambda$)s) (e.g., pure seawater, phytoplankton, colored dissolved organic matter (CDOM), and non-algal particles (NAP)) (IOCCG, 2006).

    Ocean waters are loosely classified into two categories based on the constituents present in the water and those constituents'

relationship with $R_{rs}(\lambda)$. Waters, where the IOP($\lambda$)s co-vary with the presence of phytoplankton and its derived CDOM, are generally referred to as Case I waters and are typically found offshore in the open ocean. The IOP($\lambda$)s of Case I waters are typically singularly parameterized using the concentration of the phytoplankton pigment Chlorophyll-a ($[Chla]\,[mgm^{-3}]$) and, hence, result in single parameter bio-optical models. Unlike Case I waters, Case II waters, which are most commonly found in coastal and turbid environments, consist of phytoplankton, NAP, and CDOM, none of which are ubiquitously covaried.

Following, multiple parameters are required to represent Case II water IOP($\lambda$)s. Many joint retrieval algorithms (Chowdhary et al., 2005; Hasekamp et al., 2011; Xu et al., 2016; Stamnes et al., 2018) assume single parameter bio-optical models developed



for Case I waters, whereas only a few algorithms (Chowdhary et al., 2012; Gao et al., 2018, 2019; Fan et al., 2021) adopt multi-parameter (3-7 parameters) bio-optical models. There is no universal bio-optical model for either Case I or Case II waters.

While Case I bio-optical models are frequently inadequate to represent Case II water IOP($\lambda$)s, the larger parameter space required for Case II parameterizations leads to longer forward model simulation times or decreases in the likelihood of accurate retrieval convergence. Thus, the balance between the model fidelity and the parameter space is vital to improve retrievals and uncertainties. Based on Principal Component Analysis (PCA) of $R_{rs}(\lambda)$ data from selected open and coastal waters (no inland waters were considered), it has been suggested that the number of independent free parameters a retrieval algorithm might meaningfully retrieve is roughly four in the absence of additional information or in the presence of overly restrictive measurement uncertainties (Cael et al., 2023; Hannadige et al., 2023).

The Multi-Angular Polarimetric Ocean coLor (MAPOL) joint retrieval algorithm (Gao et al., 2018, 2019, 2020) is an optimization approach that retrieves aerosol microphysical properties (aerosol optical depth (AOD), single scattering albedo (SSA), size distribution, and refractive index) and in-water properties ($R_{rs}(\lambda)$, $[Chla]$ and component $IOP(\lambda)$s) simultaneously. MAPOL is implemented with a 7-parameter bio-optical model for Case II waters and with a single-parameter model for Case I waters. Gao et al. (2019) showed that the retrieval uncertainties are partially dependent on the size of the parameter space, including that of bio-optical models, by comparing retrieval results under the two bio-optical models. In this study, we carried out MAP retrievals with the Research Scanning Polarimeter (RSP) measurements from two NASA airborne campaigns (Aerosol Characterization from Polarimeter and Lidar (ACEPOL) (https://www-air.larc.nasa.gov/missions/acepol) (Knobel-spiesse et al., 2020) and North Atlantic Aerosols and Marine Ecosystems Study (NAAMES) (https://www-air.larc.nasa.gov/missions/naames) (Behrenfeld et al., 2019). The RSP measurements were selected such that the underlying waters represent clear to turbid water conditions. The retrieval results were checked against the AOD product from MODIS and High Spectral Resolution Lidar (HSRL)-2 (Burton et al., 2013) and ocean color products ($R_{rs}(\lambda)$ and $[Chla]$) from MODIS. The retrieval uncertainties have been evaluated with respect to the Glory uncertainty requirement for AOD (Mishchenko et al., 2004) and PACE uncertainty requirements for open ocean $R_{rs}(\lambda)$ (Werdell et al., 2019).

The goal of this study is to analyze the impact of different multi-parameter bio-optical models on joint retrieval performance and their resultant uncertainties and to reduce retrieval uncertainties in the MAPOL algorithm for Case II waters. Fan et al. (2021) has studied the impact of bio-optical models on retrieval accuracy, but their results were limited to radiometric measurements under a single view angle. Based on single-pixel MAPOL retrievals, Gao et al. (2019) showed that the 7-parameter bio-optical model is more applicable over coastal waters where the single-parameter model is less robust, compared to open waters. In our study, we used 3 bio-optical models with different numbers of parameters and we evaluated the distribution of their retrieval cost function values (Sec. 3) from the ensemble retrievals. We show that the retrieval cost function distribution under 3 and 7-parameter bio-optical models are impacted by the type of water present, whereas it always results in a narrow distribution for the 1-parameter bio-optical model. Hannadige et al. (2023) showed that multi-parameter bio-optical models with 3 and 5 parameters show similar retrieval performances for semi-analytical algorithm (SAA) based $R_{rs}(\lambda)$ inversions. Here, we have shown that the retrieval performances of 3 and 7-parameter bio-optical models with the MAP joint retrieval



algorithm, MAPOL are similar to each other. Our findings suggest that the 3 and 7-parameter models are suitable to apply over both open and coastal waters whereas the 1-parameter model is less robust over coastal waters.

This study also expects to improve the performance in the POLYnomial-based Atmospheric Correction (POLYAC) algorithm (Hannadige et al., 2021) which is an AC algorithm for hyperspectral single-view radiometers that relies on collocated MAPOL
retrievals from MAPs. We believe this work would make significant impacts on the Earth science community by developing more efficient and robust retrieval algorithms for aerosols and ocean color. The bio-optical models can also be easily applied to other retrieval algorithms as well.

This paper is organized as follows. Section 2 reviews the data used in the study; Section 3 describes the MAPOL algorithm and the respective bio-optical models; Section 4 presents the methodology and the retrieval results along with an uncertainty
assessment under three different scenes; Section 5 discusses the overall results; and, finally Section 6 summarizes the conclusions.

## 2   Data

### 2.1   Airborne data

In this study, we used airborne RSP measurements acquired from the ACEPOL 2017 (https://www-air.larc.nasa.gov/missions/
acepol/index.html) (Knobelspiesse et al., 2020) and NAAMES 2015 (https://www-air.larc.nasa.gov/missions/naames/index.html) (Behrenfeld et al., 2019) airborne field campaigns. The ACEPOL campaign was held from October 19 to November 9, 2017, covering California, Nevada, Arizona, New Mexico, and the coastal Pacific Ocean. The NAAMES 2015 was the first deployment of the NAAMES campaign conducted from November 5 to December 2, 2015, over the North Atlantic Ocean.

RSP is an along-track scanner, with 152 viewing angles within $\pm 60°$. It has 9 spectral channels spanning the visible to
shortwave infrared (SWIR) with central wavelengths of each band located at 410, 470, 550, 670, 865, 960, 1590, 1880, and 2250 nm. RSP-1 and RSP-2 are two versions of the RSP instrument that differ in measurement uncertainty characterizations. RSP measurements over oceans have been used for aerosol and ocean color retrievals in multiple studies (Chowdhary et al., 2005, 2012; Stamnes et al., 2018; Gao et al., 2019, 2020) with promising performances. In the ACEPOL campaign, RSP-2 measurements were acquired with a relative radiometric characterization uncertainty of approximately 0.03 and polarimetric
characterization uncertainty of about 0.002 in Degree of Linear Polarization (DoLP), whereas in NAAMES 2015 campaign RSP-1 measurements were acquired with radiometric (relative) and polarimetric characterization uncertainties of approximately 0.015 and 0.002 respectively. The instrument noise model for RSP is provided in Knobelspiesse 2019 (Knobelspiesse et al., 2019).

We performed MAP retrievals across three RSP flight legs over selected open and coastal water regions. From the ACEPOL
campaign, we selected a coastal leg across Monterey Bay where the waters were mostly clear offshore and turbid when closer to the coast. From the NAAMES campaign, we selected a coastal leg across Delaware Bay and an open ocean leg offshore and outward from Delaware Bay. Each case has been named based on the campaign and the type of water present: ACEPOL-Mix, NAAMES-Coastal, and NAAMES-Open. Gao et al. (2019) showed a single pixel retrieval from the NAAMES-Coastal case





**Table 1.** Summary of the datasets used in this study.

| RSP Leg | ACEPOL-Mix | NAAMES-Coastal | NAAMES-Open |
|---|---|---|---|
| Date | 07 November 2017 | 4 November 2015 | 4 November 2015 |
| Number of pixels | 62 | 40 | 106 |
| UTC time range | 20:13 - 20:25 | 18:21 - 18:26 | 17:34 - 18:20 |
| Aircraft altitude | 20 km | 6.7 km | 6.8 km |
| Solar zenith angle | 53° | 59° | 55° |
| Relative azimuth angle | 75° | 110° | 75° |
| Scattering angle range | $[105°,132°]$ | $[91°,132°]$ | $[93°,133°]$ |

inside Delaware Bay comparing the retrieval performances of 1 and 7-parameter bio-optical models. The details of the three

cases are summarized in Table 1 and Figure 1. The three cases were selected based on the availability of RSP measurements in cloud-free conditions, the water turbidity of the location, and the availability of desired MODIS retrieval products. The turbidity of the waters was assumed based on MODIS $[Chla]$ retrievals (Hu et al., 2012).

RSP wavelength bands corresponding to water vapor absorption (960, and 1880 nm), as well as those wavelength bands with high noise (1590, and 2250 nm bands only for DoLP), were excluded in the retrieval. The viewing angles contaminated by

sun glint and clouds are excluded from the retrieval to reduce retrieval uncertainty. For each location of interest, 5 consecutive pixels along the RSP leg were averaged to achieve better measurement accuracy. The RSP legs with averaged pixels are shown in Figure 1. For the ACEPOL and NAAMES campaigns, the size of each averaged pixel is approximately 1 km and 0.5 km respectively. The corresponding averaged measurements (reflectance and DoLP) were applied in the retrieval.

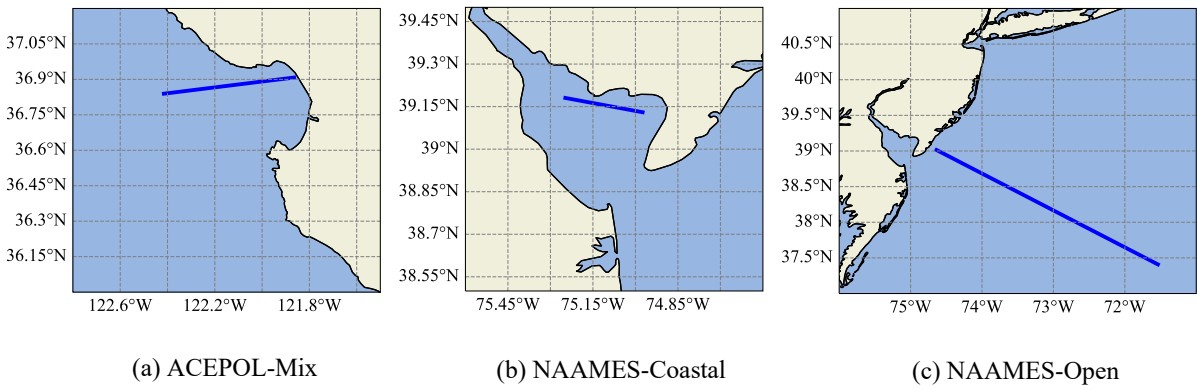

(a) ACEPOL-Mix        (b) NAAMES-Coastal        (c) NAAMES-Open

**Figure 1.** Geographical locations of the selected RSP legs.





## 2.2 Validation data

The AOD from the ACEPOL campaign is validated against HSRL-2. Due to the lack of at-sea in situ validation data, we performed sanity checks of the retrieval results using MODIS AOD and $R_{rs}(\lambda)$ products. MODIS is a single-view angle, multi-spectral imager on both the NASA Terra and Aqua satellite platforms. The MODIS-OC product (NASA Ocean Color Web, 2020 [https://oceancolor.gsfc.nasa.gov]) is processed using the standard NASA atmospheric correction algorithm (Mobley et al., 2016) developed based on the atmospheric correction algorithm (Gordon and Wang, 1994) as modified by (Ahmad et al.,

2010). We used level-2 ocean color (OC) products from the MODIS instrument on board the Aqua satellite. It provides a spatial coverage of 1km resolution at nadir. The OC products include $R_{rs}(\lambda)$ at 412, 443, 469, 488, 531, 547, 555, 645, 667, and 678 nm and $[Chla]$ via the OCI algorithm (Hu et al., 2012). The level-2 MODIS aerosol product (https://ladsweb.modaps. eosdis.nasa.gov) over the ocean and vegetated land surfaces are provided by the dark target (DT) algorithm (Levy et al., 2013). The spatial coverage has a 3 km resolution at the nadir. It provides AOD over the ocean at 7 spectral bands (470, 550, 660,

860,1240, 1630, and 2130 nm). The ACEPOL 2017 campaign flew the HSRL-2 along with RSP, the former instrument also providing accurate data for AOD validation.

## 3 The MAPOL framework

The MAPOL joint retrieval algorithm simultaneously retrieves aerosol and ocean color properties from MAP measurements. It has been validated with synthetic RSP data (Gao et al., 2018) and real RSP (Gao et al., 2019; Hannadige et al., 2021) and

SPEX airborne measurements (Gao et al., 2020; Hannadige et al., 2021).

### 3.1 Retrieval cost function

The algorithm minimizes the difference between the MAP measurements and forward model simulations for CAOS (Zhai et al., 2009, 2010). The forward model simulation is iteratively optimized (Levenberg – Marquardt non-linear least squares optimization) by perturbing the set of free parameters that represent the atmosphere and ocean optical properties. The least

squares cost function ($\chi^2(\mathbf{x})$) used to quantify the difference between the measurement and the forward model simulation is defined as,

$$\chi^2(\mathbf{x}) = \frac{1}{N}\Sigma(\lambda_i)\left(\frac{[\rho_t(i) - \rho_t^f(\mathbf{x};i)]^2}{\sigma_t^2(i)} + \frac{[P_t(i) - P_t^f(\mathbf{x};i)]^2}{\sigma_P^2(i)}\right) \qquad (1)$$

where $\rho_t = \pi r^2 L_t/\mu_0 F_0$ is the total measured reflectance and $P_t = \pi\sqrt{Q_t^2 + U_t^2}/L_t$ is the total measured DoLP. $L_t$, $Q_t$, and $U_t$ are the first three Stokes parameters measured at sensor level; $\mu_0$ is the cosine of the solar zenith angle; $F_0$ is the extraterres-

trial solar irradiance corrected for the Sun-Earth; and $r$ is the Sun-Earth distance in astronomical units. $\rho_t^f$ and $P_t^f$ denote the total reflectance and DoLP simulated from the forward model. $\mathbf{x}$ is the state vector of the retrieval; i is the measurement index corresponding to a particular angle or wavelength; and N is the total number of measurements used in the retrieval. $\sigma_t$ and $\sigma_P$ are the total uncertainties of reflectance and DoLP used in the algorithm; that includes instrument measurement uncertainties



(Knobelspiesse et al., 2019), variance due to averaging nearby pixels, and forward model uncertainties estimated as 0.015 and
0.002 for the radiometric and polarimetric uncertainties respectively (Gao et al., 2019). The uncertainty correlation between
angles has been ignored (Knobelspiesse et al., 2012; Gao et al., 2022).

The $\chi^2$ value of a converged retrieval indicates the goodness of fit of the retrieval. A $\chi^2$ value substantially larger than 1
suggests the insufficiency of the forward model to accurately represent a given set of MAP measurements. A $\chi^2$ close to 1
implies that the difference between the measurement and the corresponding forward model simulation is within the uncertainty
quantified by $\sigma_t$ and $\sigma_P$. In this study, we used $\chi^2$ values obtained under each retrieval to assess their retrieval quality and
performances.

## 3.2  Forward model

The forward model of the MAPOL algorithm is a vector radiative transfer model based on the successive order of scattering
method (Zhai et al., 2009, 2010). The CAOS system is defined as three layers: a top molecular layer, a middle layer with
mixed aerosols and molecules (2 km height), and an ocean layer bounded by a rough water surface (Cox and Munk, 1954).
The aerosol size distribution is composed of five spherical aerosol sub-modes: three fine modes and two coarse modes, each
with a log-normal distribution. The mean radius and variance are fixed (Gao et al., 2020). The complex refractive index spectra
of the two aerosol modes are based on PCA of datasets representing spectral refractive indices of water, dust-like, biomass
burning, industrial, soot, sulfate, water-soluble (Shettle and Fenn, 1979), and sea salt (de Almeida et al., 1991). The refractive
indices are approximated as $m(\lambda) = m_0 + \alpha_1 p_1(\lambda)$, where $m_0$ and $\alpha_1$ are fitting parameters, and $p_1(\lambda)$ is the first order of the
principal component.

In the MAPOL forward model, the analytical Fournier-Forand phase function ($F_p$) (Fournier and Forand, 1994) is used to
represent the particulate scattering phase function. The $F_p$ is determined by $B_p(= b_{bp}/b_p)$ (Mobley et al., 1993). The overall
phase function of water is obtained by mixing $F_p$ with that of a pure water phase function, which is then multiplied by the
normalized Mueller matrix derived from measurements (Voss and Fry, 1984; Kokhanovsky, 2003), to obtain the total Mueller
matrix of water assuming invariant polarization properties (Zhai et al., 2017).

MAPOL retrieves the spectral aerosol refractive indices described by 8 parameters (2 (fine and coarse) modes × 2 PCA × 2
parts (real and imaginary), aerosol volume densities (5 parameters, one for each aerosol sub-mode), 1 parameter to represent
the roughness of ocean surface, i.e., wind (characterized by isotropic Cox Munk model (Cox and Munk, 1954)) and either 1, 3
or 7 parameters to represent water IOP($\lambda$)s depending on the choice of bio-optical model in the retrieval.

### 3.2.1  Bio-optical models

MAPOL includes two ocean bio-optical models in the forward model to represent Case I and Case II waters separately. The
Case I water bio-optical model ("C1P1") is a single-parameter model based on $[Chla]$, where the number followed by "P"
stands for the number of free parameters in the model. The Case II ("C2P7") model contains seven bio-optical parameters. In
this study, we have included a third Case II water bio-optical model with three parameters ("C2P3"). A detailed description of
the bio-optical models is given below.



C2P7 (Eq. 2-5) is a coastal or Case II bio-optical model with 7 parameters.

$$a_{ph}(\lambda) = A_{ph}(\lambda)[\text{Chla}]^{E_{ph}(\lambda)} \tag{2}$$

$$a_{dg}(\lambda) = a_{dg}(440) \exp[-S_{dg}(\lambda - 440)] \tag{3}$$

$$b_{bp}(\lambda) = b_{bp}(660)\left(\frac{\lambda}{660}\right)^{-S_{bp}} \tag{4}$$

$$B_p(\lambda) = B_p(660)\left(\frac{\lambda}{660}\right)^{-S_{Bp}} \tag{5}$$

where $a_{ph}(\lambda)[m^{-1}]$ is the absorption coefficient of phytoplankton parameterized in terms of $[Chla]$ using $A_{ph}$ and $E_{ph}$ spectral coefficients obtained from (Bricaud et al., 1998); $a_{dg}(\lambda)[m^{-1}]$ is the spectral absorption coefficient of CDOM + NAP; $b_{bp}(\lambda)[m^{-1}]$ is the spectral backscattering coefficient of particulate matter; $B_p(\lambda)$ is the spectral backscattering fraction of particulate matter; $S_{dg}[nm^{-1}]$ is the spectral exponential slope of $a_{dg}(\lambda)$ in $nm^{-1}$; $S_{bp}$ is the spectral slope of the power law function of $b_{bp}(\lambda)$; and, $S_{Bp}$ is the spectral slope of the power law function of $B_p(\lambda)$. The 7 free parameters are $[Chla]$, $a_{dg}(440), b_{bp}(660), B_p(660), S_{dg}, S_{bp}$, and $S_{Bp}$ where 440 and 660 represent reference wavelengths in nm.

C2P3 is a 3-parameter model simplified from the C2P7 model (Eq. 2-5). To reduce the number of free parameters, we fixed the spectral slopes $S_{dg}$ at 0.018 $nm^{-1}$, $S_{bp}$ at 0.3, and $S_{Bp}$ at 0, and assumed a spectrally invariant backscattering fraction $B_p$ of 0.01 (Whitmire et al., 2007). The fixed value of $S_{bp}$ was obtained from a sensitivity analysis carried out by Hannadige et al. (2023). The remaining free parameters of the model are $[Chla]$, $a_{dg}(440)$ and, $b_{bp}(660)$.

C1P1 (Eq. 6-10) is a $[Chla]$ based single parameter Case I water bio-optical model (Zhai et al., 2015, 2017) . The absorption coefficient of phytoplankton $a_{ph}(\lambda)$ is the same as Eq. 2. The absorption $a_{dg}(\lambda)$ is given by Eq. 3 as in C2P7 model, though $S_{dg}$ is fixed at 0.018 $nm^{-1}$ and $a_{dg}(440)$ is specified by Eq. 6 and 7 in terms of $[Chla]$ (IOCCG, 2006):

$$a_{dg}(440) = p_2 a_{ph}(440) \tag{6}$$

$$p_2 = 0.3 + \frac{5.7 \times 0.5 a_{ph}(440)}{0.02 + a_{ph}(440} \tag{7}$$

Similarly, $b_{bp}(\lambda)$ is also contributed only from phytoplankton and is expressed in terms of $[Chla]$ (Huot et al., 2008).

$$b_{bp}(\lambda) = B_p \times b_p(\lambda) \tag{8}$$





where $b_p(\lambda)[m^{-1}]$ is the spectral scattering coefficient of particulate matter.

$$b_p(\lambda) = b_p(660)\left(\frac{\lambda}{660}\right)^{-S_p} \tag{9}$$


$$b_p(660) = 0.347[Chla]^{0.766} \tag{10}$$

In Eq. 9, $S_p$ is the spectral coefficient of $b_p$. For $0.02 < [Chla] < 2\ mgm^{-3}$, $S_p = -0.5(\log_{10}[Chla] - 0.3)$. For $[Chla] > 2$ $mgm^{-3}$, $S_p = 0$. $B_p$ is assumed to be spectrally invariant and is described as $B_p = [0.002 + 0.01(0.50 - 0.25\log_{10}[Chla])$. The three bio-optical models are summarized in Figure 2.

| C2P7 (7 parameters) | | C2P3 (3 parameters) |
|---|---|---|
| $a_{\mathrm{ph}}(\lambda) = A_{\mathrm{ph}}(\lambda)[\mathbf{Chla}]^{E_{\mathrm{ph}}(\lambda)}$ $a_{\mathrm{dg}}(\lambda) = \mathbf{a_{dg}(440)}\exp[-\mathbf{S_{dg}}(\lambda - 440)]$ $b_{\mathrm{bp}}(\lambda) = \mathbf{b_{bp}(660)}\left(\frac{\lambda}{660}\right)^{-\mathbf{S_{bp}}}$ $B_{\mathrm{p}}(\lambda) = \mathbf{B_p(660)}\left(\frac{\lambda}{660}\right)^{-\mathbf{S_{Bp}}}$ | | $a_{\mathrm{ph}}(\lambda) = A_{\mathrm{ph}}(\lambda)[\mathbf{Chla}]^{E_{\mathrm{ph}}(\lambda)}$ $a_{\mathrm{dg}}(\lambda) = \mathbf{a_{dg}(440)}\exp[-0.018(\lambda - 440)]$ $b_{\mathrm{bp}}(\lambda) = \mathbf{b_{bp}(660)}\left(\frac{\lambda}{660}\right)^{-0.3}$ $B_{\mathrm{p}} = 0.01$ |
| C1P1 (1 parameter) | | |
| $a_{\mathrm{ph}}(\lambda) = A_{\mathrm{ph}}(\lambda)[\mathbf{Chla}]^{E_{\mathrm{ph}}(\lambda)}$ $a_{\mathrm{dg}}(\lambda) = a_{\mathrm{dg}}(440, [\mathbf{Chla}])\exp[-0.018(\lambda - 440)]$ $b_{\mathrm{bp}}(\lambda) = b_{\mathrm{p}}(660, [\mathbf{Chla}])\left(\frac{\lambda}{660}\right)^{-S_{\mathrm{bp}}([\mathit{Chla}])}$ $B_{\mathrm{p}} = [\,0.02 + 0.01(0.5 - 0.25\log_{10}[\mathbf{Chla}])]$ | | |

**Figure 2.** The summary of MAPOL bio-optical models. The free parameters of each model are indicated in bold.

**4  Retrieval results**

We performed retrievals with the MAPOL algorithm (Sec. 3) for the 3 cases (ACEPOL-Mix, NAAMES-Coastal, and NAAMES-Open) described in Section 2. Separate retrievals were carried out using each bio-optical model (C2P7, C2P3, and C1P1 described in Sec 3.2.1) for all the cases, regardless of the type of water they represent.

The final retrieval results are based on the ensemble retrieval technique (Gao et al., 2019, 2020). The technique can reduce 265 the likelihood of convergence of the algorithm at local minima instead of the global minimum. The ensemble retrieval was carried out by performing 100 retrievals for each averaged RSP pixel. For each retrieval, the retrieval parameters are initialized with randomly generated initial values of each parameter, which are confined within a boundary as specified in Table 2 for



**Table 2.** The upper and lower boundaries of the bio-optical model parameters

| Parameter | Model | Lower/Upper boundaries |
|---|---|---|
| $[Chla](mgm^{-3})$ | C1P1, C2P3, C2P7 | 0.001/30.0 |
| $a_{dg}(440)(m^{-1})$ | C2P3, C2P7 | 0.001/2.5 |
| $S_{dg}(nm^{-1})$ | C2P7 | 0.005/0.02 |
| $b_{bp}(660)(m^{-1})$ | C2P3, C2P7 | 0.001/0.1 |
| $S_{bp}$ | C2P7 | 0.001/2.5 |
| $B_p(660)$ | C2P7 | 0.001/0.05 |
| $S_{Bp}$ | C2P7 | $-0.2/0.2$ |

bio-optical model parameters (Gao et al., 2018, 2019; Hannadige et al., 2023) and as in Gao et al. (2018) for atmospheric parameters.

The retrievals were sorted based on their $\chi^2$ distribution, which is attributed to whether the ensemble of retrievals converged at the global minimum (narrow $\chi^2$ distribution) or different local minima (broad $\chi^2$ distribution). For each of the RSP pixels, we averaged 30% (i.e. cumulative probability = 30%) of the total retrievals to calculate the final retrieval results. We studied average retrievals from all three bio-optical models using different cumulative probabilities at a time. About 30% cumulative probability yielded the lowest $\chi^2$ and retrieval variability. The selection of cumulative probability less than 30% did not leave

enough ensemble retrievals to estimate the average retrieval results. (For the C1P1 model this number is about 70%. To make it consistent across all three bio-optical models, 30% was selected). It should be noted that all the converged retrievals under the three case studies yielded $\chi^2$ larger than 0.3. The minimum and maximum $\chi^2$ values within this 30% are denoted as $\chi^2_{min}$ and $\chi^2_{max}$ respectively. For all three cases, the selection of the first 30% lowest $\chi^2$ retrievals resulted in $\chi^2_{max}$ values which are about 5 points higher than the $\chi^2_{min}$ (that is $\chi^2_{max} \approx 5 + \chi^2_{min}$). The choice of the cumulative probability or the $\chi^2_{max}$ depends

on the accuracy requirement of the retrieval.

The resultant uncertainties of the retrieval parameters are determined as the standard deviation of the retrievals within $\chi^2_{min}$ and $\chi^2_{max}$. The uncertainties are associated with different initial values in the optimization. Due to a large number of retrieval parameters and the nonlinearity of the cost functions, the choice of the initial values often becomes important (Gao et al., 2020). Gao et al. (2020) demonstrated that the uncertainty derived from ensemble retrievals within $\chi^2_{min} - \chi^2_{max}$ range is comparable

to the uncertainty calculated from the error propagation method (Knobelspiesse et al., 2012). The error propagation method directly relates the retrieval uncertainties to measurement uncertainties. The evaluation of uncertainties calculated from the error propagation method is subjected to a future study.





## 4.1 ACEPOL-Mix

The minimum retrieval cost function value $\chi^2_{min}$ is affected by the type of water present and the bio-optical model employed
in the retrieval. For relatively clear waters, where $1 < [Chla] < 3\ mgm^{-3}$, the $\chi^2_{min}$ obtained under all the three bio-optical
models are similar ($2 < \chi^2_{min} < 3$). The average $\chi^2_{min}$ value within 30% of the lowest $\chi^2$ retrievals ($\chi^2_{avg_{30\%}}$) is comparable
to the $\chi^2_{min}$ (Fig. 3). For C2P3 and C2P7 $\chi^2_{avg_{30\%}} < 1.5 \times \chi^2_{min}$. This suggests that the ensemble retrieval $\chi^2$ values have a
narrow spread attributed to the fact that most of the retrievals have reached their global minimum.

With increasing turbidity towards the coast, the $\chi^2_{min}$ values from C1P1 retrievals follow an increasing trend with increasing
$[Chla]$. Both the C2P3 and C2P7 models shows similar $\chi^2_{min}$ values along the track, whose $\chi^2_{min}$ values ($< 5$) also tend to
increase with increasing $[Chla]$ but with less variability than that of C1P1 ($\chi^2_{min} > 5$). Larger $\chi^2_{min}$ indicates the inability of
the forward model to accurately fit the MAP measurement. In other words, the C1P1 model is insufficient to fully represent the
turbid water $IOP(\lambda)$s compared to the C2P3 and C2P7 bio-optical models.

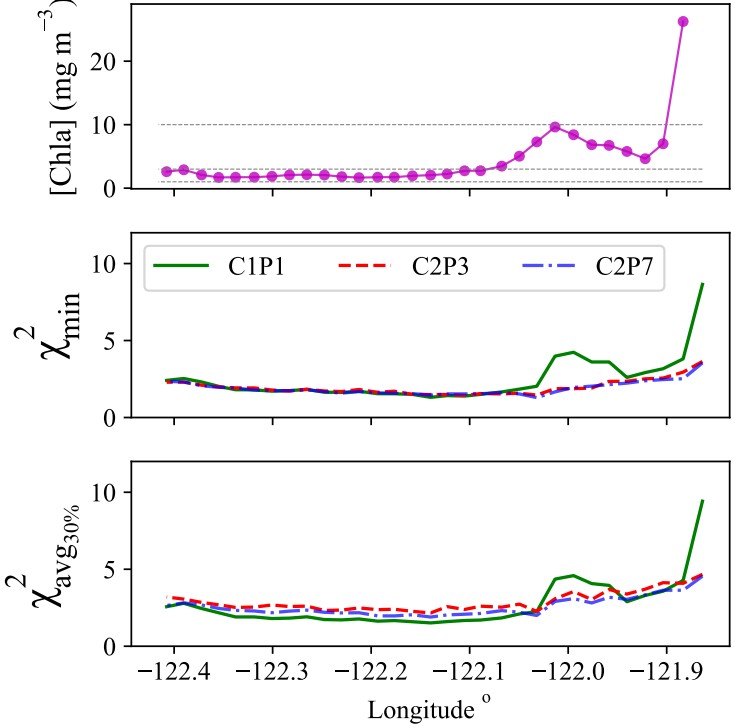

**Figure 3.** ACEPOL-Mix: The top figure shows the MODIS retrieved $[Chla]$. The gray dashed lines indicate $[Chla] = 1, 3$ and $10\ mgm^{-3}$.
The middle figure shows the $\chi^2_{min}$ obtained for the RSP retrievals across the ACEPOL-Mix leg under the three bio-optical models; C1P1,
C2P3 and, C2P7. The bottom figure shows the average $\chi^2_{min}$ value for the 30% of the lowest $\chi^2$ retrievals. Data is given with respect to the
longitude of the location. The coast is to the right-hand side of the plots





We further validated the retrieval results and evaluated the retrieval uncertainties (Figs. 4 and 6) associated with each bio-
optical model using AOD retrievals from HSRL-2 and MODIS. MODIS and HSRL-2 AOD (Fig. 5) were collocated with RSP
within a maximum distance of around 1.7 km and 0.5 km. There are no in situ $R_{rs}(\lambda)$ measurements available for validation
for this scene. Instead, we compared $R_{rs}(\lambda)$ with collocated MODIS $R_{rs}(\lambda)$ collected within a maximum distance of 0.5 km.
The time difference between MODIS and RSP measurements is roughly 1 hour. The MODIS 412, 469, 555, and 667 nm ocean
color bands were chosen to compare the corresponding RSP $R_{rs}(\lambda)$ at 410, 470, 550, and 670 nm bands and MODIS 470, 550,
and 660 nm AOD bands were chosen to compare corresponding RSP AOD at 470, 550 and 670 nm bands. In this case study,
the AOD and $R_{rs}(\lambda)$ retrievals obtained by averaging 30% of the lowest $\chi^2$ cases were compared with that obtained for the
$\chi^2_{min}$ case (The results are not shown here). The comparison of retrieved AOD at 532 nm with HSRL-2 is given in Figure 5.
For clear visualization, the density of the pixels has been reduced in the plots. The vertical bars indicate the $1\sigma$ uncertainty.

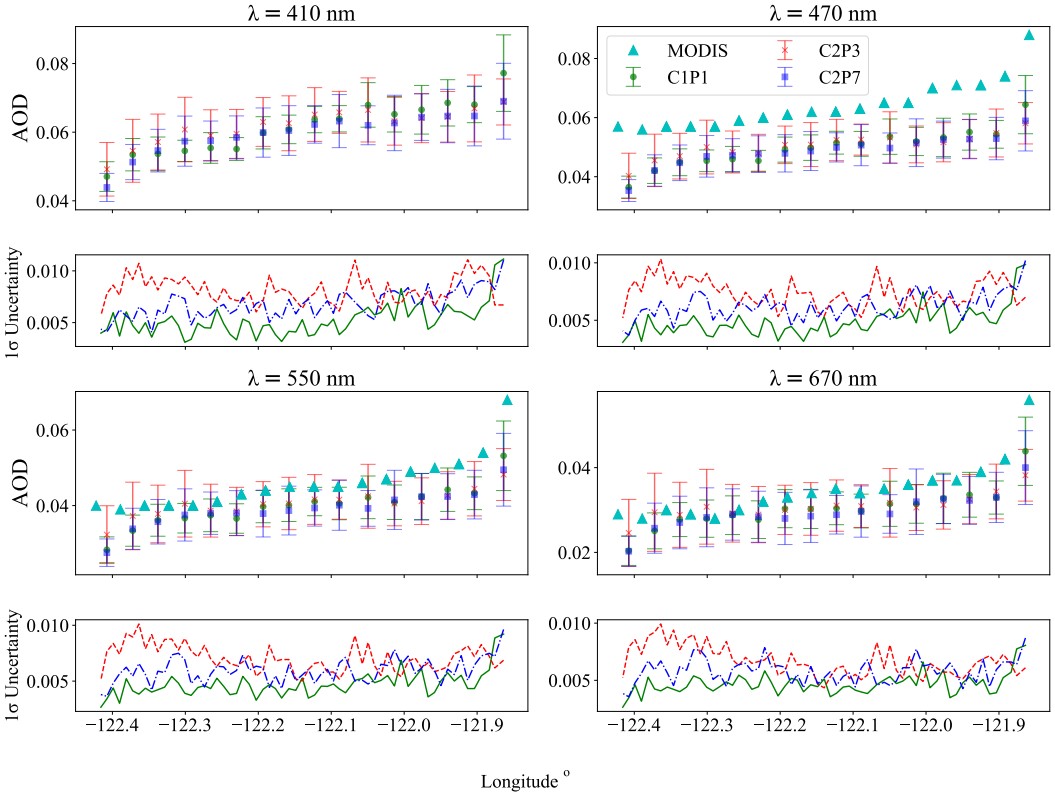

**Figure 4.** ACEPOL-Mix: The comparison of RSP retrieved averaged spectral AOD across the Monterey Bay with MODIS AOD product and
retrieval uncertainty. Results are shown for the retrievals under the three bio-optical models C1P1, C2P3, and, C2P7 at 410, 470, 550, and
670 nm for averaged retrievals. The vertical bars indicate the $1\sigma$ uncertainty. Data is given with respect to the longitude of the location. The
coast is to the right-hand side of the plots





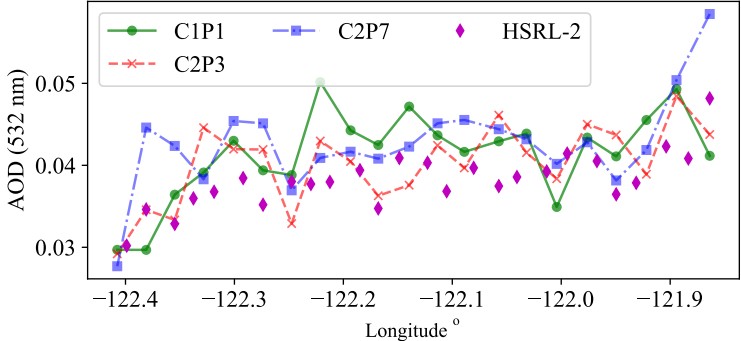

**Figure 5.** ACEPOL-Mix: The comparison of retrieved AOD at 532 nm with HSRL-2 AOD at 532 nm. The AOD obtained for the lowest $\chi^2$ case is shown here. Results are shown for the retrievals under the three bio-optical models C1P1, C2P3 and, C2P7. Data is given with respect to the longitude of the location. The coast is to the right-hand side of the plots

Regardless of the selected bio-optical model or the turbidity of the water, all three models, C1P1, C2P3, and, C2P7 show

similar AOD values, suggesting that the bio-optical model does not substantially influence AOD retrievals (Fig. 4). The comparison of AOD retrievals shows good agreement with MODIS within $1\sigma$ uncertainty limits except at 470 nm. Overall, the MODIS estimated AOD is larger than that from the MAPOL algorithm under all 3 bio-optical models. Based on the AOD retrieval comparison with respect to HSRL-2 (Fig. 5) the C2P3 model shows the best agreement among the 3 bio-optical models (Table 3). The differences between the HSRL-2 and RSP retrieved AOD may be related to different sampling volumes and

viewing geometries of the instruments.

In the comparison of $R_{rs}(\lambda)$ retrievals under the three bio-optical models (Fig. 6), MODIS shows negative $R_{rs}(\lambda)$ values at shorter wavelengths (410, and 470 nm) over the one or two pixels closest to the coast around 121.95° W. This indicates that the MODIS atmospheric correction algorithm has overestimated the aerosol signal over coastal waters. There are no negative $R_{rs}(\lambda)$ in the MAPOL retrievals. MODIS estimated $R_{rs}(\lambda)$ values are higher than those from MAPOL for relatively clear

waters at 410, 470, and 550 nm, but agree well at 670 nm with $R_{rs}(\lambda)$ retrieved from C2P3 and C2P7 models. The C1P1 model also agrees well at 670 nm, but not when closer to the coast. The differences between MODIS products and MAPOL retrievals using the 3 bio-optical models are given in Table 3.

The corresponding retrieval uncertainties for AOD and $R_{rs}(\lambda)$ are calculated as discussed in Section 4. The retrieved AOD values are similar across the 3 bio-optical models, but their AOD uncertainties differ due to the differences in their retrieval $\chi^2$

distribution. C1P1 shows the lowest AOD and $R_{rs}(\lambda)$ retrieval uncertainties. Yet, even though C1P1 shows smaller uncertainties compared to the other two models, the accuracy of the $R_{rs}(\lambda)$ retrievals is not satisfactory for the two most nearshore pixels with respect to MODIS. The average uncertainty is less than 0.01 for AOD at all the given RSP wavelengths. This falls within the AOD uncertainty requirement defined by the Glory mission, namely, a maximum of 0.02 over the ocean (Mishchenko et al., 2004). Overall, the C2P3 AOD uncertainty is slightly higher than that of C2P7. But it becomes smaller than that of C2P7 over

the coastal waters.





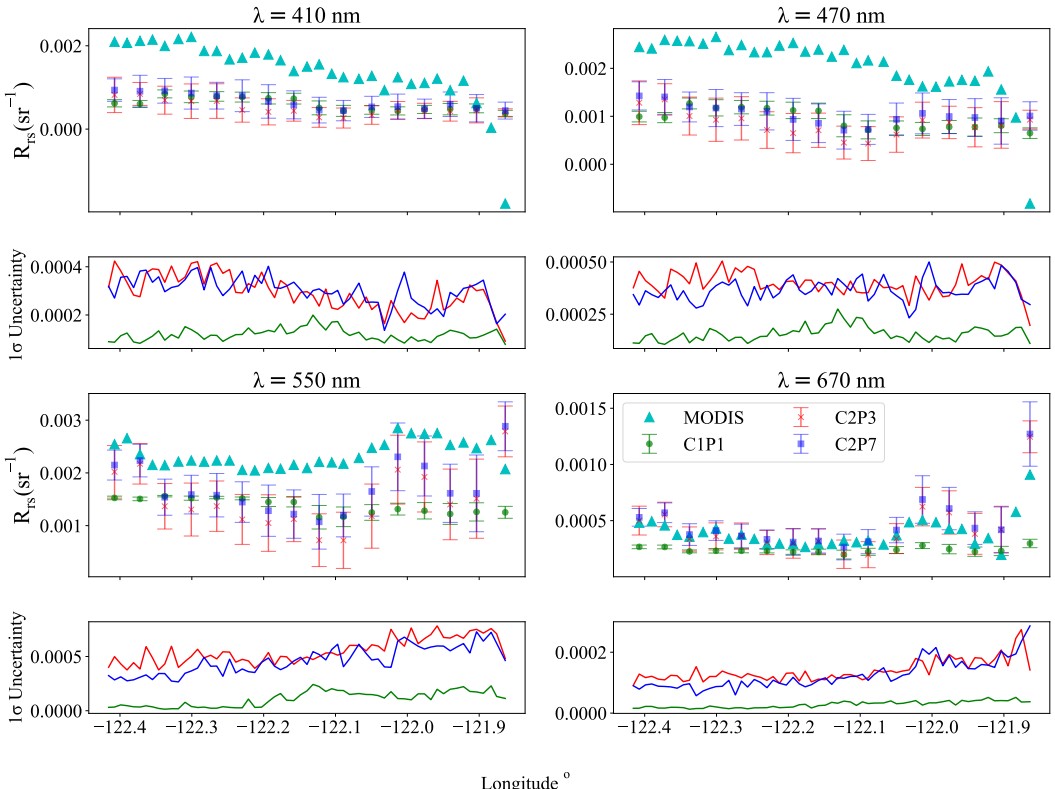

**Figure 6.** The same as Figure 4 but for $R_{rs}(\lambda)$

**Table 3.** ACEPOL-Mix: The average relative differences ($\frac{1}{N}\sum\left[\frac{a_{model}-a_{MODIS}}{a_{MODIS}}\right]$ (%), where $a$ is either AOD or $R_{rs}(\lambda)$ and $N$ is the number of total retrieved pixels) of AOD and $R_{rs}(\lambda)$ between MODIS and the 3 bio-optical models (C1P1, C2P3, and C2P7) at 410, 470, 550, and 670 nm. Negative $R_{rs}(\lambda)$ from MODIS were excluded. The differences are based on 30% averaged retrievals. The standard deviation of the relative differences is given inside the parentheses

|  |  | 410 nm | 470 nm | 550 nm | 670 nm |
|---|---|---|---|---|---|
|  | C1P1 | - | 22.2 (4.5) | 12.8 (5.6) | 12.5 (6.1) |
| AOD | C2P3 | - | 19.9 (4.8) | 9.8 (5.9) | 10.0 (6.3) |
|  | C2P7 | - | 23.3 (4.5) | 14.0 (5.3) | 13.8 (6.2) |
|  | C1P1 | 60.7 (6.0) | 56.8 (4.7) | 39.1 (9.3) | 34.6 (11.0) |
| $R_{rs}(\lambda)$ | C2P3 | 66.9 (6.9) | 61.9 (10.7) | 40.1 (15.0) | 14.9 (12.8) |
|  | C2P7 | 58.4 (5.6) | 53.6 (9.8) | 30.2 (13.0) | 17.3 (15.7) |





The $R_{rs}(\lambda)$ uncertainty from C2P3 and C2P7-based retrievals are similar with a maximum of 0.0004, 0.0005, 0.0007, and, 0.0003 $sr^{-1}$ at 410, 470, 550 and 670 nm respectively. These uncertainties fall within the PACE defined $R_{rs}(\lambda)$ uncertainty: from 400 to 600 nm the absolute uncertainty is 0.0006 $sr^{-1}$, and from 600 to 710 nm the absolute uncertainty is 0.0002 $sr^{-1}$ (Werdell et al., 2019). For C1P1 the $R_{rs}(\lambda)$ uncertainty is less than 0.0002 $sr^{-1}$ for all the wavelengths shown in Fig. 6 and falls within PACE defined $R_{rs}(\lambda)$ uncertainty.

The C1P1 AOD uncertainty is comparable with the other two models but C1P1 $R_{rs}(\lambda)$ uncertainty is significantly lower than the other two models. One reason can be explained as the total number of free parameters in the retrieval. With the C1P1 model, there is a total of 15 parameters to be retrieved. For C2P3 and C2P7 that increases to 17 and 21 respectively. With fewer parameters, it is easier to converge at the global minimum within the parameter space, or a similar local minimum is always achieved. Here, for the C1P1 model, the majority of the retrievals are converged to the same point (either a local minimum or the global minimum), hence the uncertainty is relatively small. With a larger number of free parameters in the retrieval, convergence can be achieved at a local minimum more often than at the global minimum. That makes the $\chi^2$ distribution widespread, hence the uncertainty becomes larger.

### 4.2 NAAMES-Coastal

The NAAMES-Coastal case (2015 November 04) covers RSP retrievals over Delaware Bay (Fig. 1 (b)), which is a coastal water region with high turbidity. The $\chi^2_{min}$ value obtained for each pixel with the three bio-optical models (C1P1, C2P7, and C2P3) is given at the bottom of Figure 7. The averaged $\chi^2_{min}$ for the 30% of the lowest $\chi^2$ ($\chi^2_{avg_{30\%}}$) cases is the same as $\chi^2_{min}$ for C1P1 and roughly twice the $\chi^2_{min}$ value for both C2P3 and C2P7. We did not see a significant difference between the retrieval results obtained from the lowest $\chi^2$ case and 30% average, hence only the averaged AOD and $R_{rs}(\lambda)$ retrievals are shown here. The MODIS $[Chla]$ data (Fig. 7) shows values larger than 5 $mgm^{-3}$ and the peak value exceeds 20 $mgm^{-3}$. The C1P1 model has shown the highest $\chi^2_{min}$ values around 100 still with a narrow $\chi^2$ distribution, whereas both C2P3 and C2P7 models show $\chi^2_{min}$ values around 1.5. The large $\chi^2_{min}$ values around 100 with narrow $\chi^2$ distributions imply the insufficiency of the C1P1 model to represent highly turbid coastal waters. Overall, C2P3 and C2P7 models show the same capability to represent turbid coastal waters.

We collocated MODIS AOD and ocean color products within a maximum distance of 0.8 km. The time difference between MODIS and RSP scanning times is approximately 1 hour. The MODIS 412, 469, 555, and 671 nm ocean color bands are chosen to compare $R_{rs}(\lambda)$ at 410, 470, 550, and 670 nm, and the MODIS 470, 550, and 660 nm AOD bands are chosen to compare AOD at 470, 550, and 670 nm.

The averaged AOD obtained under the C1P1 model is larger than those obtained with C2P3 and C2P7, likely because the C1P1 model misrepresents the water properly in Delaware Bay (Fig. 8). Correspondingly, the C1P1 $R_{rs}(\lambda)$ is less than that from C2P3 and C2P7 (Fig. 9). At 410 and 470 nm, the $R_{rs}(\lambda)$ retrieved with C2P7 is on average larger than that from C2P3, but similar values are retrieved at 550 and 670 nm. The MODIS $R_{rs}(\lambda)$ agrees well with C2P3 and C2P7 at 550 and 670 nm. The average relative differences between MODIS $R_{rs}(\lambda)$ and MAPOL retrievals using the 3 bio-optical models are given in Table 4. Due to the limited MODIS AOD values, average relative differences are not shown for the AOD retrievals.





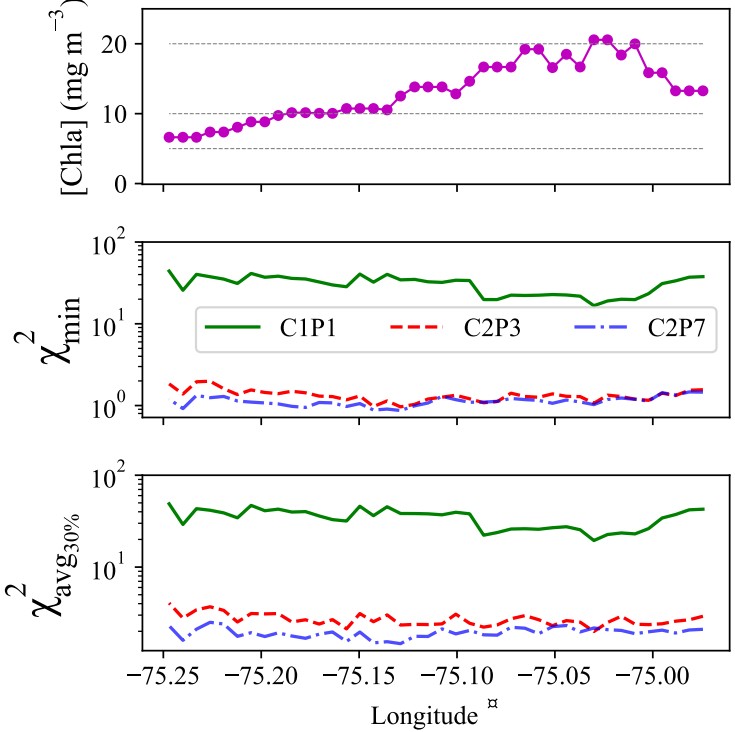

**Figure 7.** NAAMES-Coastal: The top figure shows MODIS $[Chla]$. The gray dashed lines indicate $[Chla]$ = 5, 10 and 20 $mgm^{-3}$. The middle figure shows $\chi^2_{min}$ obtained for the RSP retrievals under the three bio-optical models: C1P1, C2P3 and, C2P7. The bottom figure shows the average $\chi^2_{min}$ value for the 30% of the lowest $\chi^2$ retrievals. Data is given with respect to the longitude of the location

**Table 4.** NAAMES-Coastal: The average relative differences (%) of $R_{rs}(\lambda)$ between MODIS and the 3 bio-optical models (C1P1, C2P3, and C2P7) at 410, 470, 550 and 670 nm. The differences are based on 30% averaged retrievals. Negative $R_{rs}(\lambda)$ from MODIS were excluded. The standard deviation of the relative differences is given inside the parentheses

|      | 410 nm      | 470 nm      | 550 nm     | 670 nm     |
|------|-------------|-------------|------------|------------|
| C1P1 | 31.4 (21.2) | 63.4 (10.2) | 75.5 (4.5) | 90.5 (1.7) |
| C2P3 | 17.4 (14.7) | 18.4 (6.2)  | 7.1 (4.3)  | 7.6 (5.7)  |
| C2P7 | 60.8 (35.3) | 7.9 (4.5)   | 10.5 (5.9) | 7.9 (4.8)  |

365      The AOD and $R_{rs}(\lambda)$ retrieval uncertainties (Fig. 8 and 9) are generally similar across the three bio-optical models, with a few exceptions seen for C1P1 $R_{rs}(\lambda)$ uncertainty at longer wavelengths. The average AOD uncertainty is less than 0.02 at all the given RSP wavelengths and meets the AOD uncertainty requirement for climate models as assessed by Mischenko et al. (2004). The $R_{rs}(\lambda)$ uncertainty for the C2P7 model is larger for shorter wavelengths (410 and 470 nm), where the

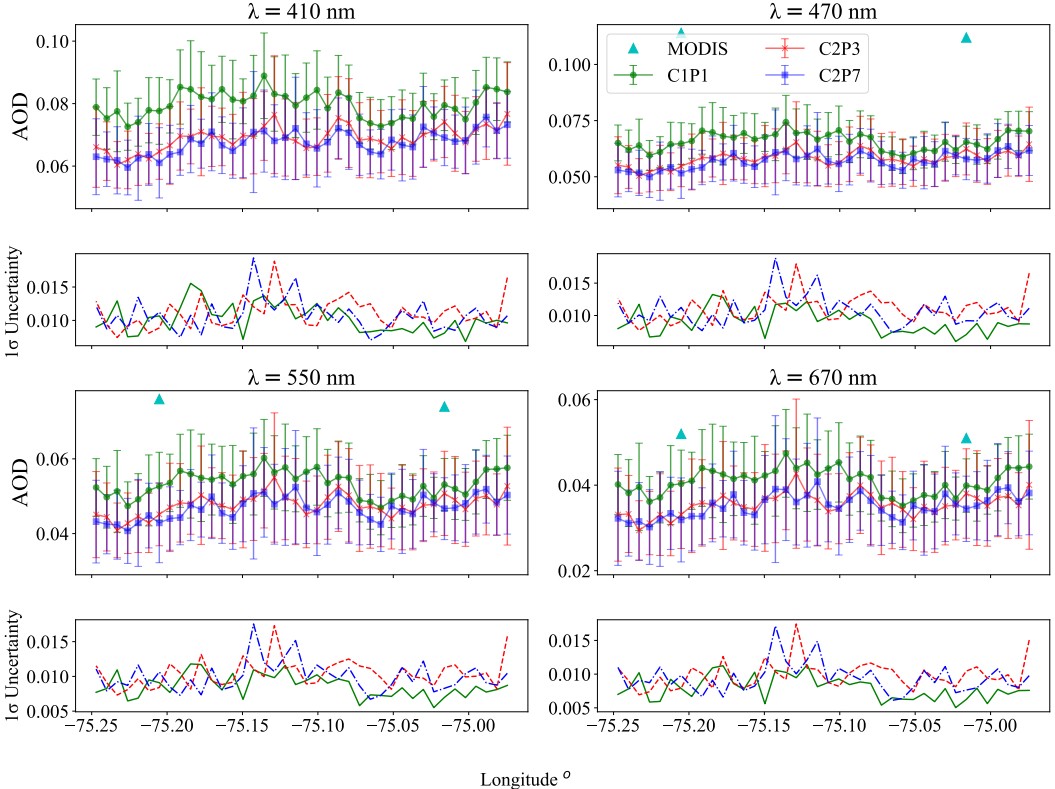

**Figure 8.** NAAMES-Coastal: The comparison of the RSP retrieved averaged AOD across the Delaware Bay with MODIS AOD product and uncertainty. Results are shown for the retrievals under the three bio-optical models C1P1, C2P3, and, C2P7 at 410, 470, 550, and 670 nm for averaged retrievals. The vertical bars indicate the $1\sigma$ uncertainty. Data is given with respect to the longitude of the location

corresponding $R_{rs}(\lambda)$ signals are small. Overall, the C2P3 and C2P7 models result in $R_{rs}(\lambda)$ uncertainties near the uncertainty

370    defined by the PACE mission except at 670 nm. Even though the $R_{rs}(\lambda)$ retrieval uncertainties are very small, the significantly larger $\chi^2$ values under the C1P1 model suggest that the C1P1 model is not suitable to represent the coastal water properties.

### 4.3    NAAMES-Open

The NAAMES-Open case (2015 November 04) covers RSP retrievals along the open ocean outward from Delaware Bay (Fig.
1 (c)). The $\chi^2_{min}$ values obtained for each pixel, under the three bio-optical models (C1P1, C2P7, and C2P3) are shown in the

middle panel of Fig. 10. The averaged $\chi^2_{min}$ for the 30% of the lowest $\chi^2$ cases is the same as $\chi^2_{min}$ for C1P1, and around 5 times the $\chi^2_{min}$ value for both C2P3 and C2P7 showing larger $\chi^2$ distributions. This implies that C2P3 and C2P7 models result in retrievals that converge at different local minima, instead of the global minimum. The MODIS $[Chla]$ values (the top panel of Fig.10) are less than 0.5 $mgm^{-3}$ in the open ocean and increase up to 4 $mgm^{-3}$ closer to the coast/Delaware Bay. The



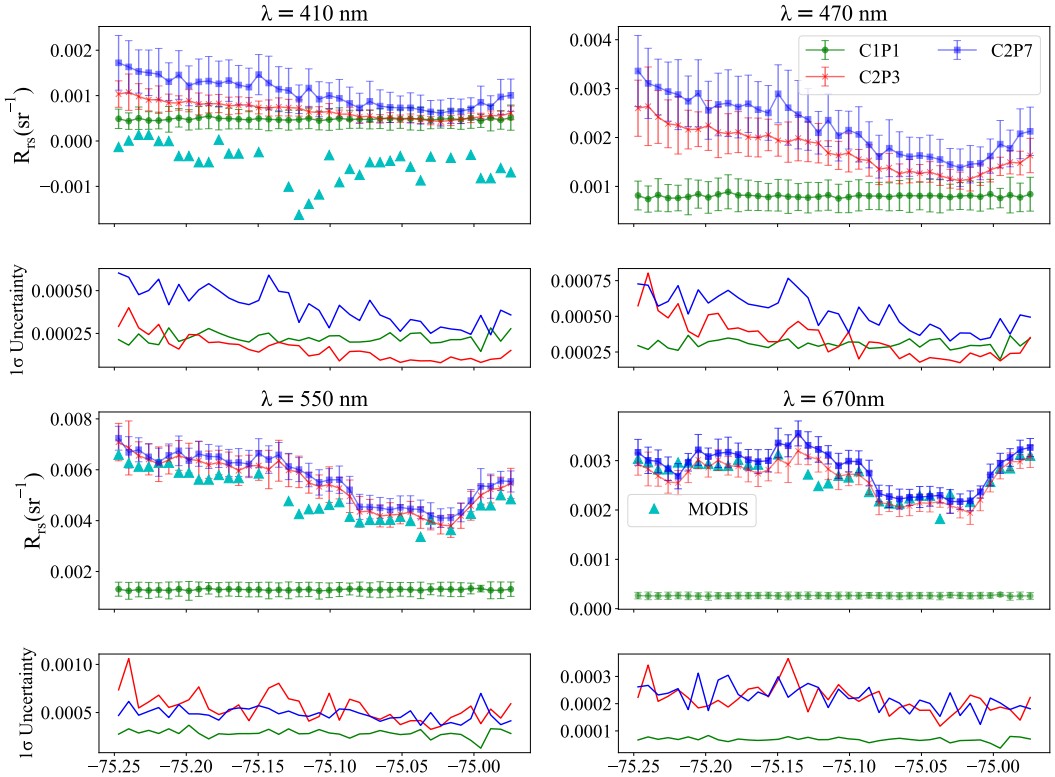

**Figure 9.** The same as Figure 8 but for $R_{rs}(\lambda)$

$\chi^2_{min}$ values are similar across all three bio-optical models with values around 1. There are some pixels from longitude 71.5°

W to 72.3° W which shows larger $\chi^2_{min}$ values which we found to be attributed to cirrus cloud contamination.

  We collocated MODIS AOD and $R_{rs}(\lambda)$ within a maximum distance of 1.4 km and 0.5 km respectively. The time difference between MODIS and RSP, scanning times is 1 hour. The MODIS 412, 469, 555, and 671 nm ocean color bands are used to compare the corresponding RSP $R_{rs}(\lambda)$ at 410, 470, 550, and 670 nm and MODIS 470, 550, and 660 nm AOD bands are used to compare corresponding RSP AOD at 470, 550, and 670 nm.

The comparison with MODIS AOD (Fig. 11) shows a better agreement with averaged C1P1 AOD. Some exceptions are seen in the locations that were attributed to cloud contamination. Unlike the previous two cases, the C1P1 averaged $R_{rs}(\lambda)$ show the best agreement with MODIS $R_{rs}(\lambda)$, mostly over open waters (Fig. 12). The C2P3 and C2P7 averaged $R_{rs}(\lambda)$ show better agreement only when closer to the coast, where C1P1 is not expected to provide a complete representation of the water optical properties. For C2P3 and C2P7 models, the comparison of AOD and $R_{rs}(\lambda)$ retrievals obtained for the lowest $\chi^2$ retrieval of

the ensemble retrieval, show better agreement with MODIS AOD and MODIS $R_{rs}(\lambda)$ compared to the averaged retrievals. The relative differences between MODIS and MAPOL retrieved AOD corresponding to $\chi^2_{min}$ and $\chi^{avg_{30\%}}$ are given in Table 5 and the same for $R_{rs}(\lambda)$ is given in Table 6. There is a significant difference seen in the relative difference values between



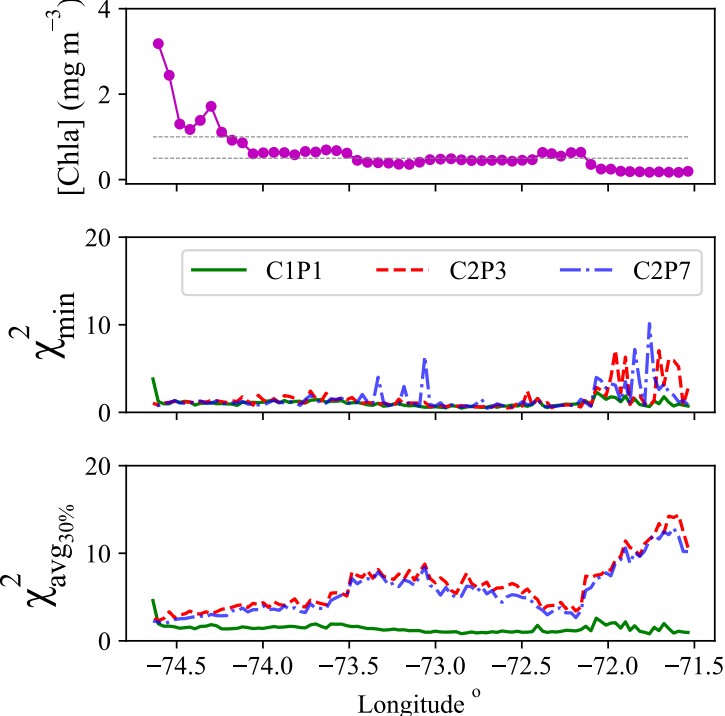

**Figure 10.** NAAMES-Open: The top figure shows MODIS $[Chla]$. The gray dashed lines indicate $[Chla]$=0.5 and 1 $mgm^{-3}$. The middle figure shows $\chi^2_{min}$ obtained for the RSP retrievals under the three bio-optical models: C1P1, C2P3 and, C2P7. The bottom figure shows the average $\chi^2_{min}$ value for the 30% of the lowest $\chi^2$ retrievals. Data is given with respect to the longitude of the location

$\chi^2_{min}$ and $\chi_{avg_{30\%}}$ for $R_{rs}(\lambda)$ which is not evident for AOD. The distribution of $\chi^2$ values in the ensemble retrieval therefore largely affects $R_{rs}(\lambda)$ retrievals.

The AOD uncertainties (Fig. 11) are similar across the three bio-optical models with a maximum of 0.015 at all given wavelengths. For $R_{rs}(\lambda)$ (Fig. 12) C1P1 shows the lowest uncertainties owing to its better representation in the open ocean and small parameter space, which leads to better convergence near the global minimum. The multi-parameter models show comparably larger $R_{rs}(\lambda)$ uncertainties that are still within the PACE-defined uncertainties except at 410 nm.

## 5   Discussion

In this study, we have evaluated the retrieval performances of 3 bio-optical models within CAOSs under different water conditions. For the ACEPOL-Mix case, the waters are moderately turbid with $[Chla]$ values ranging from $3 - 20\ mgm^{-3}$. The NAAMES-Coastal case includes RSP measurements over highly turbid waters ($5 < [Chla] < 20\ mgm^{-3}$). For the NAAMES-Open case, the waters are mostly clear and become turbid when closer to the coast ($0.1 < [Chla] < 3\ mgm^{-3}$).

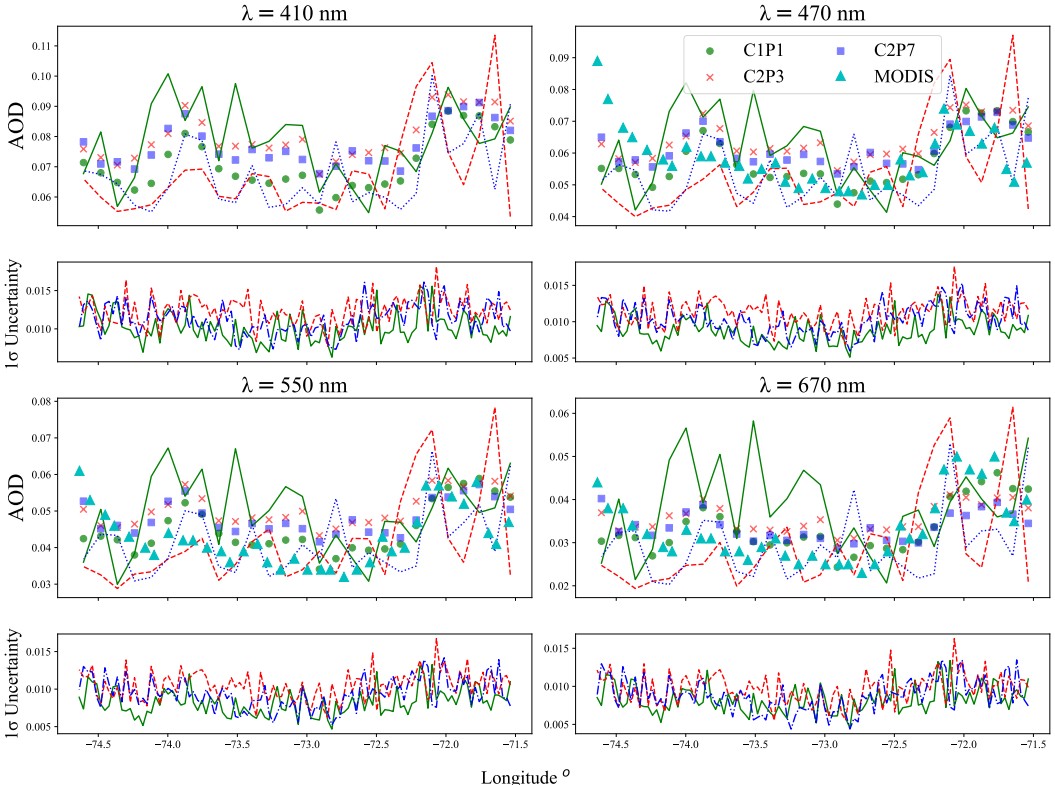

**Figure 11.** NAAMES-Open: The comparison of the RSP retrieved spectral AOD across the open ocean with MODIS AOD product and uncertainty. Results are shown for the retrievals under the three bio-optical models C1P1, C2P3, and, C2P7 at 410, 469, 554, and 670 nm for averaged retrievals. The lines (C1P1-solid, C2P3-dashed, C2P7-dotted) indicate the retrievals obtained for the $\chi^2_{min}$ case. The markers show the average retrieval. The uncertainty plots show the $1\sigma$ uncertainty for averaged retrievals. Data is given with respect to the longitude of the location. The coast is to the left-hand side of the plots

We have evaluated the retrieval performances based on the retrieval cost function values, ensemble cost function distribution, the retrieved of AOD and $R_{rs}(\lambda)$ values, and corresponding retrieval uncertainties. For the NAAMES-Open case, the C1P1 model shows low $\chi^2_{min}$ values indicating good fitting against RSP measurements. The C2P3 and C2P7 models also show good fitting with the RSP measurements, but only when the $\chi^2_{min}$ cases are considered. The C1P1 shows the best agreement in AOD and $R_{rs}(\lambda)$ retrieval results with independent data sources from the MODIS. The C1P1 retrieval performance in the ACEPOL-Mix case is satisfactory when the waters are clear ($[Chla]$ 1.5 $mgm^{-3}$), that is, towards the open ocean. The C2P3 and C2P7 models in the NAAMES-Coastal case and nearshore ACEPOL-Mix pixels show better agreement in AOD and $R_{rs}(\lambda)$ retrievals with uncertainties within the Glory uncertainty requirement for AOD and the PACE uncertainty requirement for $R_{rs}(\lambda)$.

The overall results indicate that the choice of bio-optical model (either a single parameter or multi-parameter) affects the accuracy of the retrievals, which is especially true for $R_{rs}(\lambda)$ retrievals. The retrieval performances of the C2P3 and C2P7





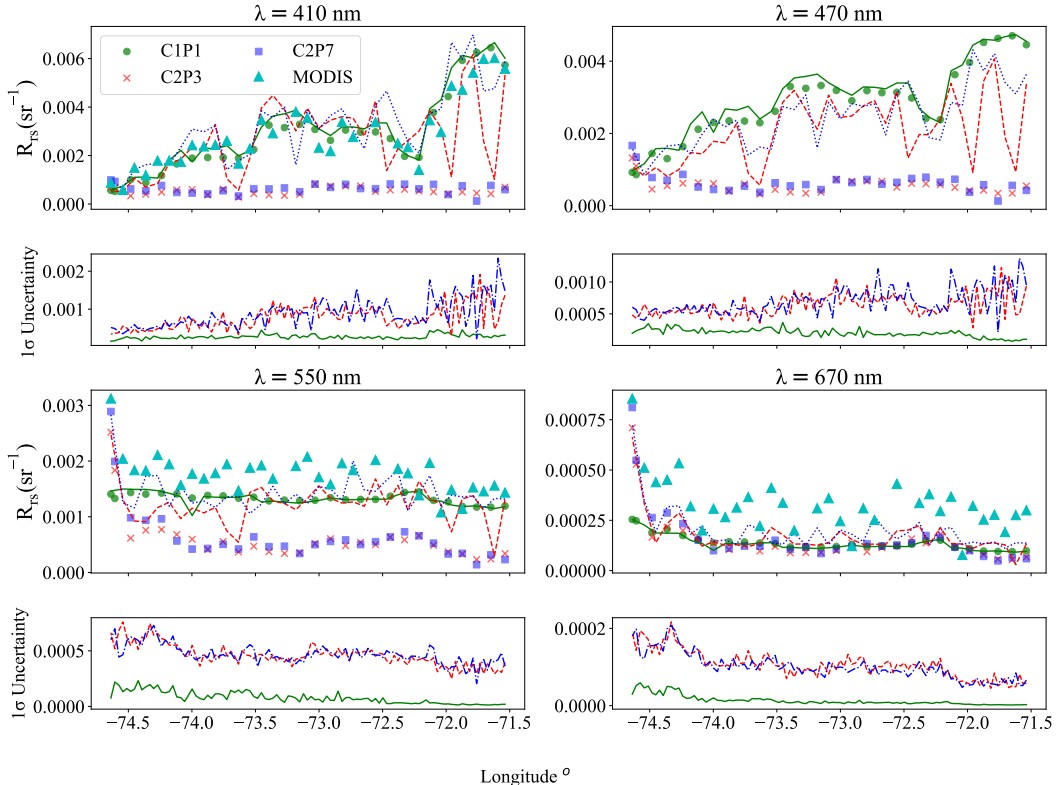

**Figure 12.** The same as Figure 11 but for $R_{rs}(\lambda)$

**Table 5.** NAAMES-Open: The average relative differences (%) of AOD between MODIS and the 3 bio-optical models (C1P1, C2P3, and C2P7) at 470, 550, and 670 nm. The differences are given for the retrievals from $\chi^2_{min}$ case and averaged retrievals $\chi^2_{avg_{30}}$%. The standard deviation of the relative differences is given inside the parentheses

|  |  | 470 nm | 550 nm | 670 nm |
|---|---|---|---|---|
| | C1P1 | 11.5 (10.5) | 13.2 (8.9) | 12.2 (7.8) |
| $\chi^2_{avg_{30\%}}$ | C2P3 | 16.2 (9.9) | 21.9 (13.0) | 16.5 (10.9) |
| | C2P7 | 13.2 (8.8) | 18.0 (11.0) | 13.5 (8.5) |
| | C1P1 | 20.3 (16.5) | 26.1 (24.0) | 32.4 (31.2) |
| $\chi^2_{min}$ | C2P3 | 17.6 (14.2) | 17.3 (14.4) | 23.3 (14.0) |
| | C2P7 | 17.7 (12.3) | . 17.4 (13.0) | 22.0 (14.3) |

models are mostly similar. For coastal waters, it is more challenging to retrieve $R_{rs}(\lambda)$ accurately due to the complex water





**Table 6.** The same as 5 but for $R_{rs}(\lambda)$

|  |  | 410 nm | 470 nm | 550 nm | 670 nm |
|---|---|---|---|---|---|
| $\chi^2_{avg_{30\%}}$ | C1P1 | 27.0 (16.6) | 25.7 (11.4) | 21.0 (8.9) | 19.2 (6.1) |
|  | C2P3 | 84.0 (7.3) | 84.4 (8.4) | 69.0 (10.6) | 52.5 (10.7) |
|  | C2P7 | 80.0 (10.4) | 81.8 (10.6) | 67.2 (12.9) | 49.7 (13.2) |
| $\chi^2_{min}$ | C1P1 | 20.6 (16.4) | 20.9 (11.4) | 21.5 (9.0) | 51.0 (6.6) |
|  | C2P3 | 27.2 (22.7) | 42.8 (15.7) | 24.8 (16.6) | 36.8 (15.6) |
|  | C2P7 | 22.3 (20.3) | 37.7 (15.2) | 21.3 (16.4) | 33.2 (16.9) |

properties that require multi-parameter bio-optical models. The C2P3 and C2P7 models show good retrieval performances over turbid waters.

We have also evaluated the distribution of ensemble $\chi^2$ values based on $\chi^2_{min}$ and $\chi^2_{avg_{30\%}}$ values. For C1P1 model the $\chi^2$ distribution from all three cases is narrow, even with larger $\chi^2$ values. This implies its ability to achieve similar convergence even if the global minimum is not reached. For C2P3 and C2P7, over moderately to highly turbid waters (ACEPOL-Mix and NAAMES-Coastal), the $\chi^2$ values are mostly closer to 1 and the distribution is nearly narrow, implying their capability to reach near the global minimum with multiple parameters over coastal waters. But in the NAAMES-Open case, C2P3 and C2P7 show widespread $\chi^2$ distributions implying their inability to reach the global minimum with multiple parameters over open waters. This can be explained by the degrees of freedom in the water leaving signal and the number of optimization parameters in the bio-optical models.

In the NAAMES-Open case, even though the averaged retrieval results from C2P3 and C2P7 are not satisfactory on average over clear waters, the retrieval results corresponding to the lowest $\chi^2$ show good agreement with MODIS AOD and $R_{rs}(\lambda)$. This implies that the C2P3 and C2P7 models still can accurately represent clear water optical properties with proper interpretation and conscientious use of the $\chi^2$ distributions. However, the averaged retrieval results differ significantly as the retrieval $\chi^2$ distributions under C2P3 and C2P7 models are widespread compared to that of C1P1. For the practical use of these bio-optical models, we suggest performing initial retrievals using the C1P1 bio-optical model and then reperforming the retrievals with either C2P3 or C2P7 models in case the C1P1 model results in significantly larger $\chi^2$ values.

The C2P3 and C2P7 models show similar retrieval performances for all three case studies. The MAPOL retrievals with the C2P3 model use 17 retrieval parameters whereas the C2P7 model uses 21 parameters. MAPOL is computationally demanding as it needs to iteratively run the radiative transfer forward model for CAOS. The time taken for a single retrieval is proportional to the size of the retrieval parameters. For the C2P3 model, it takes an average of 3 hours for a single CPU core to process one-pixel retrieval with RSP measurements whereas for the C2P7 model, the time increases up to 8 hours. Therefore, the C2P3 model is more efficient for the MAPOL algorithm to represent Case II waters.

The operational version of MAPOL, called FastMAPOL, replaces the radiative transfer forward model with neural networks, which can process several pixels within a second in a single CPU (Gao et al., 2021). We expect to update both MAPOL and





FastMAPOL algorithms with the C2P3 model in the future. The fixed parameters in the 3-parameter C2P3 model might not be true for all the water which is subject to fine-tuning. The availability of airborne MAP measurements over the oceans under cloud-free conditions is limited that we cannot cover a larger range of atmosphere and water conditions in this study. The unavailability of accurate in-situ measurements over the selected locations for the validation is yet another limitation. We expect to further improve our bio-optical models based on the MAP measurements to be acquired from the PACE mission plan to launch in early 2024.

The $[Chla]$ alone does not fully represent the turbidity of the water as the sediment/NAP concentration and CDOM availability are also important factors. There is no clear boundary between Case I and Case II waters (IOCCG, 2000), hence we cannot provide a clear set of conditions where we need to apply each of the bio-optical models used in this study. There is no universal bio-optical model to represent water bio-optical properties (Fan et al., 2021). At least two separate bio-optical models are required to represent Case I and Case II waters. The three cases in this study do not cover in-land/lake waters. The applicability of C2P3 and C2P7 to lakes or in-land waters is subject to a future study.

## 6   Conclusions

In this paper, we have evaluated the performance of the MAPOL joint retrieval algorithm under three bio-optical models. The RSP measurements from different field campaigns covering different water types are used. The retrieval performance evaluation is based on the cost function value ($\chi^2$) and the cost function distribution of the retrievals, retrieved AOD, and $R_{rs}(\lambda)$ and their uncertainty analysis. The three bio-optical models include C1P1, a single parameter Case I water model, C2P3, and C2P7, multi-parameter Case II bio-optical models. Three cases; ACEPOL-Mix, NAAMES-Costal, and NAAMES-Open, were selected based on their location and water turbidity observed with respect to $[Chla]$ derived from MODIS single-view radiometer. The NAAMES-Costal covers highly turbid waters, ACEPOL-Mix covers moderately turbid and clear waters and NAAMES-Open covers open clear waters. The retrieved AOD was validated against that from HSRL-2 (ACEPOL-Coastal) and/or MODIS and $R_{rs}(\lambda)$ was compared against that from MODIS. The MODIS $R_{rs}(\lambda)$ over turbid waters show negative values for shorter wavelengths (410 and 470 nm), hence that cannot be used as a validation dataset. On the other hand, the MODIS data products are used as a sanity check of the RSP retrievals.

We evaluated the retrieval cost function distribution of the ensemble retrievals of the three bio-optical models. The C1P1 model showed narrow $\chi^2$ distribution regardless of the type of water present or the $\chi^2_{min}$ values. The C2P3 and C2P7 models showed the widest distributions over open waters with $\chi^2_{min}$ comparable to that of C1P1. C2P3 and C2P7 showed narrow $\chi^2$ distributions over moderately to highly turbid waters with small $\chi^2$ values. These observations implied the ability of the multi-parameter bio-optical model-based retrievals to converge near the global minimum over different waters.

We also observed that the retrieval accuracies of AOD and $R_{rs}(\lambda)$ are directly related to the choice of the bio-optical model (single or multi-parameter) in the retrieval. The $R_{rs}(\lambda)$ retrieval is significantly affected. The C1P1 model shows good retrieval performances only over very clear waters ($[Chla] < 1\ mgm^{-3}$). The results suggested that the multi-parameter models, C2P3 and C2P7 are better at representing turbid coastal waters. Regardless of the retrieval technique, the C2P3 and C2P7 models



have the potential to accurately represent clear open waters (NAAMES-Open) with a conscientious interpretation of their $\chi^2$ distributions. The larger spread of $\chi^2$ values in the ensemble retrievals diminishes the ability of multi-parameter models to

accurately retrieve clear waters.

Similar to the SAA based $R_{rs}(\lambda)$ inversions (Hannadige et al., 2023), multi-parameter models (C2P3 and C2P7) also show similar retrieval performances when used with MAP joint retrieval algorithms. Based on the number of parameters present in the bio-optical model, the C2P3 model is more computationally efficient than the C2P7 model.

*Data availability.* The data files for RSP, and HSRL-2 used in this study are listed below. The RSP data are available at the NASA GISS

website https://data.giss.nasa.gov/pub/rsp. The HSRL-2 data are available from the ACEPOL website (https://www-air.larc.nasa.gov/cgi-bin/ArcView/acepol)

– ACEPOL-Mix (07 November 2017):
   RSP : RSP2-ER2_L1C-RSPCOL-CollocatedRadiances_20171107T201415Z_V003-20210305T085047Z.h5
   HSRL-2 : ACEPOL-HSRL2_ER2_20171107_R3.h5

– NAAMES-Coastal (04 November 2015):
   RSP: RSP1-C130_L1C-RSPCOL-CollocatedRadiances_20151104T182046Z_V003-20210728T201227Z.h5

– NAAMES-Open (04 November 2015):
   RSP: RSP1-C130_L1C-RSPCOL-CollocatedRadiances_20151104T173447Z_V003-20210728T201253Z.h5

*Author contributions.* NH formulated methodology and software used in this paper, performed formal analysis, investigation, data curation,

and visualization given in this paper, and wrote the original manuscript. P-WZ formulated the original concept for this study. MG and P-WZ developed the MAPOL retrieval algorithm. YH contributed to the development of the radiative transfer forward model and advised in the retrieval algorithm design. PJW advised and contributed to bio-optical models and ocean water properties. BC provided RSP measurements. KK advised on the retrieval uncertainty evaluation. All authors provided advice on the methodology and participated in writing, reviewing, and editing this paper.

*Competing interests.* The authors declare that they have no conflict of interest.

*Acknowledgements.* The authors would like to thank the ACEPOL and NAAMES teams for conducting the field campaigns and providing the data.

The hardware used in the computational studies is part of the UMBC High-Performance Computing Facility (HPCF). The facility is supported by the U.S. National Science Foundation through the MRI program and the SCREMS program, with additional substantial support from the

University of Maryland, Baltimore County (UMBC). See hpcf.umbc.edu for more information on HPCF and the projects using its resources.





*Financial support.* This project was supported by National Aeronautics and Space Administration grant 80NSSC20M0227 and Joint Center of Earth Systems Technology (JCET), University of Maryland Baltimore County (UMBC) under the JCET/GESTAR-II Graduate Fellowship Program





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
