# Peer review of "Performance evaluation of three bio-optical models in aerosol and ocean color joint retrievals"

_Atmospheric Measurement Techniques, 2023_

## Referee Comment (RC1)

**General comments**

This paper evaluated the impact of three different ocean bio-optical models on retrieval accuracy, based on the Multi-Angular Polarimetric Ocean coLor (MAPOL) joint retrieval algorithm and data from the airborne Research Scanning Polarimeter (RSP). The average differences between the MAPOL retrievals and MODIS products were reported, and the applicability of bio-optical models along geographically varying waters were evaluated.

In general, the subject of this study is suitable for AMT, but the research goal and the innovation are not clear. The retrieval performance with 1, 3 and 7-parameter bio-optical models were validated, and the findings showed that the 3 and 7-parameter models were suitable to apply over both open and coastal waters whereas the 1-parameter model was less robust over coastal waters. The findings are similar to the results in Gao et al (2019) and Hannadige et al (2023). Compared with Gao et al (2019), the 3-parameter bio-optical model was added, but the physical reason for the similar performance between 3 and 7-parameter models were not be explained. Also, the physical meaning of the model parameters, and their relationship with retrieval parameters and water properties can be explained, which helps to better analyze the impact of different models on retrieval performance. I therefore recommend major revision.

**Specific comments**

● Although the research contents have been compared with previous literatures in lines 110-122, the research subject corresponding to the study contents and the innovation of this study are not clear. In addition, the results of this study may not be presented in the introduction. The key issues and the methods used in this study can be included, to better explain the research aims.

● Line 125-127: I don't think this work can make significant impacts on the Earth science community by developing more efficient and robust retrieval algorithms for aerosols and ocean color. The joint retrieval algorithm was proposed and developed in Gao et al (2019), and this work mainly evaluated three bio-optical models in the retrieval algorithm. It is important to exactly describe the goal and significance of the study in the introduction.

● Section 3: The section 3 describes the MAPOL joint retrieval algorithm, which has been developed in Gao et al (2019). The heading of "the MAPOL framework" is usually used for algorithm development. In section 3, the retrieval algorithm can be briefly

introduced, with a focus on describing the three models to be evaluated.

● Line 198-220: Since the prior assumed uncertainties of $\sigma_t$ and $\sigma_p$ directly influence the $\chi^2$ value, I recommend elaborating the setting and calculation of $\sigma_t$ and $\sigma_p$ in the study.

● Line 241-242: The difference between the 3 and 7-parameter models is important for model evaluation. Compared with 7-parameter model, four parameters are set as fixed values. It is better to explain why these parameters are set to this value.

● The physical meanings and influencing factors of the parameters such as Sdg, Sbp, and SBp in different models can be explained. This could help to analyze the potential retrieval effects of different models.

● Figures in this paper: To increase the readability of the figure, the letters can be provided in the subfigures.

● Figure 4: Is there no MODIS product on the figure of $\lambda = 410 nm$ ? Similar phenomena appear in Figures 8 and 9 ( $\lambda = 470 nm$ ).

● In the ACEPOL-Mix case that is with moderately turbid waters, the retrieval precision of C1P1 model is good, and the uncertainties are small. This seems to be inconsistent with the conclusion that the C1P1 model is successful over extremely clear waters and unsuitable for turbid waters.

● For the performance comparison tables including tables 3-6, I recommend adding a description of the sample size.

**Technical corrections**

Language: I am not a native speaker, so take my comments with some caution.

● Line 13: are → were ?

● Line 40-43: It is better to give a brief description of the algorithm not just the algorithm used by MODIS, and the possible reason for the uncertainty.

● Line 50: what is the meaning of " the estimation of aerosols is thus important for aerosol retrievals" ?

● Line 61-62: What kind of efforts? What's the problem?

● Line 160: are excluded → were excluded ?

● Line 193-195: Does the uncertainty refer to the variance? How to determine the forward model uncertainty, and the pixel average variance?

● Line 210: How to fit the parameters?

● Line 365: Figure 8 shows that the retrieval results with C1P1 model agree well with the MODIS product for the NAAMES-Coastal case with highly turbid waters, which is not consistent with the conclusion.

● Figure 12: There is no MODIS product in the subfigure of $\lambda$=470 nm, but the table 6 presents the relative difference between MODIS and 3 bio-optical models at 470 nm.

● Line 466: widest → wider ?

---

## Author Comment (AC1)

**Referee 01**

**General comments**

This paper evaluated the impact of three different ocean bio-optical models on retrieval accuracy, based on the Multi-Angular Polarimetric Ocean coLor (MAPOL) joint retrieval algorithm and data from the airborne Research Scanning Polarimeter (RSP). The average differences between the MAPOL retrievals and MODIS products were reported, and the applicability of bio-optical models along geographically varying waters were evaluated.

In general, the subject of this study is suitable for AMT, but the research goal and the innovation are not clear. The retrieval performance with 1, 3 and 7-parameter bio- optical models were validated, and the findings showed that the 3 and 7-parameter models were suitable to apply over both open and coastal waters whereas the 1- parameter model was less robust over coastal waters. The findings are similar to the results in Gao et al (2019) and Hannadige et al (2023). Compared with Gao et al (2019), the 3-parameter bio-optical model was added, but the physical reason for the similar performance between 3 and 7-parameter models were not be explained. Also, the physical meaning of the model parameters, and their relationship with retrieval parameters and water properties can be explained, which helps to better analyze the impact of different models on retrieval performance. I therefore recommend major revision.

**AC to general comments:**

We appreciate your critical review which prompts us to further clarify the novelty of our manuscript. The main novelty of this work is to evaluate the performance of ocean bio-optical models with different free parameters in joint aerosol-ocean color retrieval algorithms and to provide recommendations on the optimal representation of the optical properties of coastal ocean waters. Though we carried out the research using MAPOL, the conclusions are applicable to any other joint retrieval algorithms of atmosphere and ocean systems, which makes its impact far-reaching in the Earth science community.

As the reviewer correctly pointed out Gao et al. (2019) have partially touched on this subject by comparing the 1-parameter open water model and the 7-parameter coastal water model with MAPOL using multi-angle polarimeter (MAP) data. They concluded that the choice of bio-optical model affects the retrieval accuracy, and the 7-parameter model is more suitable for the coastal waters than the Case I model. What was not covered is whether seven parameters are necessary to represent coastal waters. In other words, are there other bio-optical models of coastal ocean water that have similar or superior performance? Though it sounds trivial, this question is indeed important to explore as the number of free parameters directly changes the number of forward model and Jacobian simulations, which as a consequence greatly affects the speed and stability of the least square fitting algorithms. Bio-optical models with fewer free parameters with superior or similar performance are highly desired, which makes it easier to develop operational algorithms for satellite sensors.

The reviewer is also right that the work by Hannadige et al. (2023) is highly relevant as they use principal component (PC) analysis to show that 3 PCs can be used to represent the spectral

variation of a large number of remote sensing reflectance Rrs data. The origin of the Rrs data is from both synthetic simulation and in-situ measurements representing diverse conditions of global ocean waters. Hannadige et al. (2023) also showed that the multi-parameter models (3-parameter vs 7 parameters in combination with different parameterization schemes) have similar retrieval performances in the Rrs inversions with a semi-analytic algorithm, regardless of their number of free parameters. This demonstrated that the use of a 3-parameter bio-optical model is feasible. Note that Hannadige et al. (2023) used in-situ Rrs data. This has never been demonstrated in inverting airborne MAP measurements.

In this work, we extend the work by Gao et al. (2019) and Hannadige et al. (2023) and study the performance of different bio-optical models in joint retrieval algorithms of airborne MAP measurements, which is also applicable to satellite sensors. We confirmed that a simplified bio-optical model of 3 parameters performs equally well as the original 7-parameter model. In addition, we also revealed a number of important features which have not been published before. They include:

1.  The 1-parameter model uncertainty is small, even when it fails to converge over coastal waters. This suggests that using the spread of the cost function to study the uncertainty of retrieval parameters is misleading. One has to ensure convergence first.
2.  The multi-parameter (3 or 7) models can be used to represent open waters in joint retrieval algorithms, though the retrieval algorithm tends to converge to local minima which made the interpretation of the retrieval results difficult. It is preferable to develop a screening algorithm to divide open and coastal waters before performing MAP retrievals.
3.  The 3-parameter model performs equally well with the 7-parameter model, which makes it preferable as fewer free parameters lead to significantly less processing time and more stable retrieval performance.

Based on the novelty we summarized above, we believe this work is innovative, impactful, and publishable. We have updated our manuscript by rewriting part of the abstract and introduction to better reflect the novelty. For details, please refer to the revised manuscript with tracked changes. We also address the following questions to further clarify this paper's goal, novelty, and clarity.

**General comments**

1.  The physical reason for the similar performance between 3 and 7-parameter models was not explained.

    As Hannadige et al. (2023) have demonstrated only 3 independent PCs are sufficient to represent the spectra variation of Rrs, which means that the free parameters which can be used to represent the optical properties are likely to be around 3, in other words, less than 7. This explains that the 3-parameter model works in joint retrieval algorithms. The 7-parameter model provides larger parameter space, which can be used as it encompasses all the possible parameter value combinations of the 3-parameter model. The 7-parameter model is much slower to converge as more parameters lead to more forward model and Jacobian evaluations, which are very time-consuming. We have revised the manuscript (lines 451-453, here after all line numbers in AC refer to the revised manuscript) to reflect this explanation:

"The C2P3 and C2P7 models show similar retrieval performances for all three case studies. The MAPOL retrievals under the C2P3 model use 17 retrieval parameters whereas the C2P7 model uses 21 parameters. The C2P7 provides a larger parameter space that encompasses all the possible parameter value combinations of the C2P3 model, hence their performances are similar."

2. The physical meaning of the model parameters and their relationship with retrieval parameters and water properties can be explained.

The physical meaning of the model parameters is explained in Sec. 3.2.1 and summarized in Fig. 2 with the free parameters shown in bold.

**Specific comments**

1. Although the research contents have been compared with previous literature in lines 110-122, the research subject corresponding to the study contents and the innovation of this study are not clear. In addition, the results of this study may not be presented in the introduction. The key issues and the methods used in this study can be included, to better explain the research aims.

In order to clarify these, the paragraph mentioned above was updated as follows (line 92-104) and any conclusions drawn from the study were excluded from the introduction.

"The goal of this study is to examine the overall impact of bio-optical models with different numbers of free parameters on the performance and uncertainty of joint retrieval algorithms for Case II waters. Hannadige et al. (2023) showed that multiparameter bio-optical models with 3 and 5 parameters show similar retrieval performances for the semi-analytical algorithm (SAA) based on 95 in-situ multi-band Rrs measurements. An independent study showed that the number of free parameters a retrieval algorithm might meaningfully retrieve is roughly four based on in-situ hyperspectral Rrs measurements (Cael et al., 2023). Here, for the first time, we have examined to which extent these conclusions hold for the joint retrieval algorithms using airborne MAP measurements, which have not been studied before. The quality of the retrievals in this study is evaluated with respect to the magnitude of the retrieval cost function values, the distribution of retrieval cost function values (Sec. 3) from the ensemble retrievals, and the sanity check with MODIS retrievals. We studied the uncertainty of the different bio-optical models based on the spread of ensemble retrieval cost function values which is important to understand the impact of the bio-optical models on the convergence behavior of the non-linear least squares fitting algorithms. This has not been examined in previous studies. Given the inherent problems associated with MODIS retrievals over optically complex scenes, we consider the MODIS products as merely a reference, rather than a validation dataset."

2. Line 125-127: I don't think this work can make significant impacts on the Earth science community by developing more efficient and robust retrieval algorithms for aerosols and ocean color. The joint retrieval algorithm was proposed and developed in Gao et al (2019), and this work mainly evaluated three bio-optical models in the retrieval algorithm. It is important to exactly describe the goal and significance of the study in the introduction.

We have addressed the novelty issue earlier in response to the general comments. Here we re-iterate that identifying the optimal number of free parameters in joint retrieval algorithms is

important due to its profound impacts on the processing time and retrieval uncertainty. The significance of this study is, that it can provide the community with some guidelines on the use of bio-optical models in MAP joint retrieval algorithms over coastal or optically complex waters, which has not been fully studied to this extent before. The reduction of the bio-optical model parameter space can lead to reduced computational time, which is crucial for the operational use of the joint retrieval algorithms.

The following paragraphs (line 119-128) were updated to make the message clearer on the significance of this work.

"The conclusions from this study can be used to provide recommendations for selecting suitable bio-optical models for joint retrieval algorithms over coastal waters to improve their accuracy and computational efficiency. The large parameter space required for Case II parameterizations leads to longer forward model simulation times or decreases in the likelihood of accurate retrieval convergence. Thus, the balance between the model fidelity and the parameter space is vital to improve retrievals and uncertainties. This study also expects to improve the performance of the POLYnomial-based Atmospheric Correction (POLYAC) algorithm (Hannadige et al., 2021) which is an AC algorithm for hyperspectral single-view radiometers applied over optically complex scenes, such as over coastal waters. POLYAC relies on collocated MAP retrievals from the MAPOL algorithm to estimate the hyperspectral path radiance to calculate hyperspectral Rrs which is crucial for retrieving phytoplankton functional types (IOCCG, 2014). Though this study was carried out with MAPOL, the conclusions are equally applicable to other joint retrieval algorithms of aerosols and ocean color, which thus have greater impacts beyond MAPOL."

3.  Section 3: The section 3 describes the MAPOL joint retrieval algorithm, which has been developed in Gao et al (2019). The heading of "the MAPOL framework" is usually used for algorithm development. In section 3, the retrieval algorithm can be briefly introduced, with a focus on describing the three models to be evaluated.

The heading has been updated to "The MAPOL joint retrieval algorithm" and more justification has been provided for the choice of different parameter values in the bio-optical models.

4.  Line 198-220: Since the prior assumed uncertainties of $\sigma$t and $\sigma$p directly influence the $\chi$2 value, I recommend elaborating the setting and calculation of $\sigma t$ and $\sigma$p in the study.

The uncertainties are based on the instrument uncertainties characterized via the radiometric and polarimetric characterization of the RSP instruments. The RSP instrument uncertainty model is explained in Knobelspiesse et al. (2019). Forward model uncertainties are explained by Gao et al. (2021). To clarify this the following paragraph (line 192-195) has been updated.

"σt and σP are the total uncertainties of reflectance and DoLP which include the RSP instrument characterization (Knobelspiesse et al., 2019), variance due to averaging nearby pixels, and forward model uncertainties estimated as 0.015 and 0.002 for the radiometric and polarimetric uncertainties respectively (Gao et al., 2022). The uncertainty correlation between angles has been ignored (Knobelspiesse et al., 2012; Gao et al., 2022)."

5. Line 241-242: The difference between the 3 and 7-parameter models is important for model evaluation. Compared with 7-parameter model, four parameters are set as fixed values. It is better to explain why these parameters are set to this value.

The following paragraph (line 242-250) has been updated to provide more explanation of how the parameter values are determined.

"C2P3 is a 3-parameter model simplified from the C2P7 model (Eq. 2-5). To reduce the number of free parameters, we fixed the spectral slopes. Sdg typically varies between 0.01 and 0.02 nm-1 in natural waters. Based on the in-situ measurements over oceans (Roesler et al., 1989) most of the existing bio-optical models such as Default Configuration Generalized IOP (GIOP-DC) model (Werdell et al., 2013) adopt Sdg=0.018 nm-1. It has been found that the particulate backscattering ratio from in-situ measurements shows little or no spectral dependence and the mean particulate backscattering ratio is 0.010 (Chami et al., 2005; Whitmire et al., 2007). We have fixed SBp at 0 and assumed a spectrally invariant backscattering fraction Bp of 0.01. Sbp typically varies between 0 and 2 from small to large particles (Werdell et al., 2013). Sbp was fixed at 0.3 in this study which was obtained by a sensitivity analysis carried out by Hannadige et al., (2023). We acknowledge that these fixed values could deviate under specific water conditions. The remaining free parameters of the model are [Chla], adg (440) and, bbp(660)."

6. The physical meanings and influencing factors of the parameters such as Sdg, Sbp, and SBp in different models can be explained. This could help to analyze the potential retrieval effects of different models.

The values of Sdg, Sbp, and SBp depend on the composition and particle size of oceanic particles. How are they impacted by the microphysical properties of the oceanic particles is beyond the scope of this study. The following lines (line 238-240) were added to the manuscript.

"The magnitude of the spectral slopes, Sdg, Sbp, and SBp depends on the composition and the size of the oceanic particles and therefore represent microphysical properties such as refractive index, effective radius, and particle size distribution slope (Jonasz, 2007)."

7. Figures in this paper: To increase the readability of the figure, the letters can be provided in the subfigures.

The legends of the figures were updated accordingly.

8. Figure 4: Is there no MODIS product on the figure of λ= 410nm? Similar phenomena appear in Figures 8 and 9 ( λ= 470nm ).

The NAAMES-Coastal case Rrs plots from the pre-print version of the manuscript include Rrs data from an older version of satellite data, which does not include Rrs at 470 nm. We have updated all the Rrs figures with up-to-date MODIS-OC data. The AOD data were from MODIS dark target (DT) algorithm, which does not include 410 nm. To make it easier to interpret AOD and Rrs retrievals we have replaced the MODIS DT AOD data with those from the OBPG atmospheric correction algorithm in the revised manuscript. The discussion and conclusion have been updated accordingly. The main conclusion on RSP retrieval remains the same.

9. In the ACEPOL-Mix case that is with moderately turbid waters, the retrieval precision of C1P1 model is good, and the uncertainties are small. This seems to be inconsistent with the conclusion that the C1P1 model is successful over extremely clear waters and unsuitable for turbid waters.

In the ACEPOL-MIX case for longitude <-122.1, 1<[chla]<3 mgm-3. That is moderately turbid towards the open ocean. When [chla]>3 mgm-3 the cost function values from the C1P1 model are larger (compared to C2P3 and C2P7), implying the fitting between the measurements and the forward model is not quite good. Even though the retrieval precision is good, our first criterion for evaluating the model's effectiveness is the cost function value. For the NAAMES-Coastal case, [chla]>5 mgm-3. The following lines in discussion and conclusion have been updated to,

(line 424-425) "The C1P1 retrieval performance in the ACEPOL-Mix case is satisfactory when the waters are relatively clear ([Chla]<3 $mgm-3), that is, towards the open ocean."

(line 496-497) "The $Rrs(\lambda)$ retrieval is significantly affected. The C1P1 model shows good retrieval performances only over relatively clear waters ([Chla] < 3 mgm−3)."

10. For the performance comparison tables including tables 3-6, I recommend adding a description of the sample size.

All the tables were updated with the sample sizes.

**Technical corrections**

11. Line 40-43: It is better to give a brief description of the algorithm not just the algorithm used by MODIS, and the possible reason for the uncertainty.

The MODIS algorithms are not the focus of this study. We mentioned MODIS to show the progression of the evolution/progression of satellite sensors. The following paragraphs were modified to provide the justification (line 40-42):

"Some of the traditional retrieval algorithms such as those for MODIS-like instruments result in larger aerosol and ocean color retrieval uncertainties when compared with the accuracy required for climate modeling (Remer et al., 2005; Sayer et al., 2016), which is due to the limited information content in single-viewing spectrometer measurements (Mishchenko et al., 2004)."

12. Line 50 : What is the meaning of " the estimation of aerosols is thus important for aerosol retrievals" ?

We agree that this sentence is vague, and it has been removed.

13. Line 61-62: What kind of efforts? What's the problem?

The problem is the negative Rrs retrievals from MODIS over coastal waters which is mainly caused by the breakdown of the black pixel assumption in the AC process and underestimation of the aerosol effects due to the impacts of absorbing aerosols in the retrieval process.

The sentences are updated to (line 59-63):

"The heritage algorithm implemented by NASA's Ocean Biology Processing Group (OBPG; https://oceancolor.gsfc.nasa.gov) works well over open waters but can produce negative Rrs($\lambda$) in blue wavelengths over turbid waters (Bailey et al., 2010) given the aforementioned reasons. Efforts have been made to overcome negative Rrs($\lambda$) (Bailey et al., 2010; He et al., 2012; Fan et al., 2021; Ibrahim et al., 2019) though the problem has not been fully resolved yet."

14. Line 193-195: Does the uncertainty refer to the variance? How to determine the forward model uncertainty, and the pixel average variance?

There are two ways to quantify the retrieval uncertainty. One is based on the theoretical formulation described in Rodgers (2001) and the other is based on the variability of retrieved parameters based on retrievals with multiple initial values. The second method is termed as the Monte Carlo error propagation method (MCEP) (Gao, M., Knobelspiesse, K., Franz, B. A., Zhai, P.-W., Cairns, B., Xu, X., and Martins, J. V.: The impact and estimation of uncertainty correlation for multi-angle polarimetric remote sensing of aerosols and ocean color, Atmos. Meas. Tech., 16, 2067–2087, https://doi.org/10.5194/amt-16-2067-2023, 2023.)

In this work, we used MCEP to quantify the uncertainty, i.e., the AOD and Rrs uncertainties are based on the 1 standard deviation of the retrieved parameter in an ensemble of retrievals.

15. Line 210: How to fit the parameters?

The reconstruction of refractive indices using PCA is based on $m(\lambda) = m_0 + \alpha_1 p_1 (\lambda)$. $p_1 (\lambda)$ is obtained from a principal analysis of a database of aerosol refractive indices. $m_0$ and $\alpha_1$ are two retrieval parameters in the joint retrieval algorithm that are obtained through the retrieval optimization process.

16. Figure 8 shows that the retrieval results with C1P1 model agree well with the MODIS product for the NAAMES-Coastal case with highly turbid waters, which is not consistent with the conclusion.

In the revised version we have used all the MODIS AOD from the atmospheric correction algorithm for better consistency between AOD and Rrs retrieval. Most of the MODIS AOD data now fall within the uncertainty limits from the 3 bio-optical models. However, the cost function values under the C1P1 model are extremely large which indicates convergence failure. This is clear in Fig. 9, in which the C1P1 Rrs is way off the other models, as well as the MODIS data.

17. Figure 12: There is no MODIS product in the subfigure of $\lambda$=470 nm, but the table 6 presents the relative difference between MODIS and 3 bio-optical models at 470 nm.

We have updated all the Rrs figures with up-to-date MODIS-OC data. All the figures have been updated to incorporate MODIS Rrs at 470 nm.

---

## Author Comment (AC2)

**Referee 02**

This paper evaluates 1-, 3-, and 7-parameter ocean bio-optical models using the Multi-Angular Polarimetric Ocean coLor (MAPOL) joint retrieval algorithm applied to data from the Research Scanning Polarimeter (RSP) in different types of water (clear and open water and turbid coastal water). The authors compare MAPOL retrievals to MODIS products.

The topic of this paper is interesting and the approach of this paper seems sound. However, it is unclear where the novelty lies in this paper. No new algorithms are developed, and the conclusion that the 3- and 7-parameter models are more versatile than the 1-parameter model is fairly intuitive. Furthermore, there are deviations between the MAPOL retrievals and MODIS retrievals which are not fully explained. Thus, I recommend acceptance with major revisions. I suggest that the authors further analyze the differences between MAPOL retrievals and MODIS retrievals and that the authors revise the paper to clarify the major novel contributions of this paper to the reader.

In general, I agree with the comments posted by Anonymous Referee #1 as RC1.

**AC to general comments:**

We appreciate your critical review which prompts us to further clarify the novelty of our manuscript. The main novelty of this work is to evaluate the performance of ocean bio-optical models with different free parameters in joint aerosol-ocean color retrieval algorithms and to provide recommendations on the optimal representation of the optical properties of coastal ocean waters. Though we carried out the research using MAPOL, the conclusions are applicable to any other joint retrieval algorithms of atmosphere and ocean systems, which makes its impact far-reaching in the Earth science community.

Gao et al. (2019) have partially touched on this subject by comparing the 1-parameter open water model and the 7-parameter coastal water model with MAPOL using multi-angle polarimeter (MAP) data. They concluded that the choice of bio-optical model affects the retrieval accuracy, and the 7-parameter model is more suitable for the coastal waters than the Case I model. What was not covered is whether seven parameters are necessary to represent coastal waters. In other words, are there other bio-optical models of coastal ocean water that have similar or superior performance? Though it sounds trivial, this question is indeed important to explore as the number of free parameters directly changes the number of forward model and Jacobian simulations, which as a consequence greatly affects the speed and stability of the least square fitting algorithms. Bio-optical models with fewer free parameters with superior or similar performance are highly desired, which makes it easier to develop operational algorithms for satellite sensors.

Hannadige et al. (2023) also showed that the multi-parameter models (3-parameter vs 7 parameters in combination with different parameterization schemes) have similar retrieval performances in the Rrs inversions with a semi-analytic algorithm, regardless of their number of free parameters. This demonstrated that the use of a 3-parameter bio-optical model is feasible. Note that Hannadige et al. (2023) used in-situ Rrs data. This has never been demonstrated in inverting airborne MAP measurements.

In this work, we extend the work by Gao et al. (2019) and Hannadige et al. (2023) and study the performance of different bio-optical models in joint retrieval algorithms of airborne MAP measurements, which is also applicable to satellite sensors. We confirmed that a simplified bio-optical model of 3 parameters performs equally well as the original 7-parameter model. In addition, we also revealed a number of important features which have not been published before. They include:

1. The 1-parameter model uncertainty is small, even when it fails to converge over coastal waters. This suggests that using the spread of the cost function to study the uncertainty of retrieval parameters is misleading. One has to ensure convergence first.
2. The multi-parameter (3 or 7) models can be used to represent open waters in joint retrieval algorithms, though the retrieval algorithm tends to converge to local minima which made the interpretation of the retrieval results difficult. It is preferable to develop a screening algorithm to divide open and coastal waters before performing MAP retrievals.
3. The 3-parameter model performs equally well with the 7-parameter model, which makes it preferable as fewer free parameters lead to significantly less processing time and more stable retrieval performance.

Based on the novelty we summarized above, we believe this work is innovative, impactful, and publishable. We have updated our manuscript by rewriting part of the abstract and introduction to better reflect the novelty. For details, please refer to the revised manuscript with tracked changes. We also address the following questions to further clarify this paper's goal, novelty, and clarity.

In order to clarify the novelty and the goal of this work, the following paragraph (line 92-104, hereafter all line numbers are referred to in the revised manuscript) has been updated.

"The goal of this study is to examine the overall impact of bio-optical models with different numbers of free parameters on the performance and uncertainty of joint retrieval algorithms for Case II waters. Hannadige et al. (2023) showed that multiparameter bio-optical models with 3 and 5 parameters show similar retrieval performances for the semi-analytical algorithm (SAA) based on 95 in-situ multi-band Rrs measurements. An independent study showed that the number of free parameters a retrieval algorithm might meaningfully retrieve is roughly four based on in-situ hyperspectral Rrs measurements (Cael et al., 2023). Here, for the first time, we have examined to which extent these conclusions hold for the joint retrieval algorithms using airborne MAP measurements, which have not been studied before. The quality of the retrievals in this study is evaluated with respect to the magnitude of the retrieval cost function values, the distribution of retrieval cost function values (Sec. 3) from the ensemble retrievals, and the sanity check with MODIS retrievals. We studied the uncertainty of the different bio-optical models based on the spread of ensemble retrieval cost function values which is important to understand the impact of the bio-optical models on the convergence behavior of the non-linear least squares fitting algorithms. This has not been examined in previous studies. Given the inherent problems associated with MODIS retrievals over optically complex scenes, we consider the MODIS products as merely a reference, rather than a validation dataset."

**Comments**

1. Figure 4: At a wavelength of 470 nm, it seems like the MAPOL retrievals significantly underestimate AOD when compared to MODIS. Generally, MODIS retrievals are outside of the error bars for MAPOL retrievals. The authors note this (in lines 310-311) but do not explain it -- what are the potential causes? This also appears to be an issue at 550 nm and 670 nm, but at these wavelengths, MODIS retrievals are at least generally within 1-sigma error bars for MAPOL retrievals.

In this pre-print version, we used MODIS dark target (DT) aerosol data product. In the revised version we replaced the DT AOD with those from the atmospheric correction algorithm of ocean color, for a better interpretation of the consistency between AOD and remote sensing reflectance Rrs. The MODIS AOD and Rrs values we have shown in the plots are relatable to each other. Based on the OBPG AC algorithm, larger AOD values lead to smaller Rrs values (which is similar to the MAPOL algorithm as well). The differences in MODIS and MAPOL results are attributed to the differences in measurement capabilities and algorithm differences. Please refer to Figure 5, which shows the comparison of AOD at 532 nm with MODIS and HSRL-2, which are mostly similar.

2. Figure 5: It might improve clarity to include a line on the plot showing where the coast actually is, or make the x-axis the distance from coast rather than longitude. At first glance, it can be difficult to understand where the coast is.

Each figure caption states where the coast is in each plot (to the right or to the left). For the NAAMES-Coastal case, the RSP leg is located along the coast of Delaware Bay. We added a sentence to the figure caption to indicate this. Using distance from the coast as the x-axis instead of the longitude can reduce the readability of the plots. To make it consistent throughout the manuscript we chose to use longitude as the x-axis.

3. Figure 6: I recommend including the full figure caption (and see above comment about deviations from MODIS).

All the figure captions are updated following your recommendation.

4. Figure 9: At 410 nm, spectral remote sensing reflectance from MODIS is consistently lower than from MAPOL (and is sometimes negative). Why is this?

Based on the updated AOD figures, the MODIS AOD at 410 nm is larger than that from MAPOL. The larger the AOD the smaller the Rrs is. Hence MODIS Rrs is smaller than that from MAPOL. Due to the underlying assumptions in the AC process of NASA OBPG algorithm (black pixel assumption), MODIS Rrs can become negative over turbid (coastal) waters.

5. Figure 9: Please include the full figure caption.

All the figure captions are updated.

6. General note on figures: I recommend including the full figure caption for all figures.

All the figure captions are updated.

7. Lines 123-124: I recommend further explanation of POLYAC and this paper's relevance to POLYAC. Perhaps POLYAC performance improvements can be used to motivate this paper (if it can be shown that novel contributions in this paper improve MAPOL retrievals).

POLYAC is an AC scheme for hyperspectral radiometers which can be used as an alternate scheme for heritage AC algorithms to derive hyperspectral Rrs when optically complex scenes such as coastal waters are involved. Accurate hyperspectral Rrs is required for ocean color studies such as for differentiation of phytoplankton functional types. POLYAC utilizes collocated MAP retrievals from MAPOL to estimate hyperspectral path radiance where its accuracy directly depends on the accuracy of the water-leaving signal estimated from MAPOL. We added the following paragraph (line 123-127) to better reflect the impact of this work to POLYAC.

"This study also expects to improve the performance of the POLYnomial-based Atmospheric Correction (POLYAC) algorithm (Hannadige et al., 2021) which is an AC algorithm for hyperspectral single-view radiometers applied over optically complex scenes, such as over coastal waters. POLYAC relies on collocated MAP retrievals from the MAPOLalgorithm to estimate the hyperspectral path radiance to calculate hyperspectral Rrs which is crucial for retrieving phytoplankton functional types (IOCCG, 2014)."

8. Lines 316-322: Why does MODIS estimate higher spectral remote sensing reflectance than MAPOL for pixels other than the ones closest to the coast?

The differences in MODIS and MAPOL results are attributed to the differences in measurement capabilities and algorithm differences. In this case, the estimation of AOD from the AC algorithm is higher than the MAPOL retrieval (Fig. 4), which leads to higher Rrs. In the first version, this was not clear as the MODIS dark target data were used, which are not consistent with the Rrs data. In the revised manuscript, we updated all AOD data from the AC algorithm so that all MODIS remote sensing data are physically consistent.

9. Line 327: The authors note that the uncertainty for AOD is less than 0.01 at all wavelengths -- but is the uncertainty adequately capturing error/differences from MODIS retrievals (especially noting the differences between MODIS and MAPOL)?

Based on the updated AOD figure, MODIS AOD from most of the cases falls within the uncertainty of the 3 bio-optical models. A small difference is seen for the NAAMES-Coastal case (Fig 8) at 410 nm, where the MODIS AOD only falls within C1P1 uncertainty. Given the overall performances, the difference between MODIS AOD and the uncertainty from C2P3 and C2P7 are insignificant.